# PMQ-VE: Progressive Multi-Frame Quantization for Video Enhancement

**ZhanFeng Feng**[1†], **Long Peng**[1†‡], **Xin Di**[1†], **Yong Guo**[2], **Wenbo Li**[3], **Yulun Zhang**[4],
**Renjing Pei**[5*], **Yang Wang**[1,6*], **Yang Cao**[1*], **Zheng-Jun Zha**[1]

[1]USTC, [2]Max Planck Institute, [3]CUHK, [4]SJTU,
[5]Institute of Automation, Chinese Academy of Sciences, [6]Chang'an University

{xiaobigfeng, longp2001, dx9826}@mail.ustc.edu.cn, ywang120@chd.edu.cn

## Abstract

Multi-frame video enhancement tasks aim to improve the spatial and temporal resolution and quality of video sequences by leveraging temporal information from multiple frames, which are widely used in streaming video processing, surveillance, and generation. Although numerous Transformer-based enhancement methods have achieved impressive performance, their computational and memory demands hinder deployment on edge devices. Quantization offers a practical solution by reducing the bit-width of weights and activations to improve efficiency. However, directly applying existing quantization methods to video enhancement tasks often leads to significant performance degradation and loss of fine details. This stems from two limitations: (a) inability to allocate varying representational capacity across frames, which results in suboptimal dynamic range adaptation; (b) over-reliance on full-precision teachers, which limits the learning of low-bit student models. To tackle these challenges, we propose a novel quantization method for video enhancement: Progressive Multi-Frame Quantization for Video Enhancement (PMQ-VE). This framework features a coarse-to-fine two-stage process: Backtracking-based Multi-Frame Quantization (BMFQ) and Progressive Multi-Teacher Distillation (PMTD). BMFQ utilizes a percentile-based initialization and iterative search with pruning and backtracking for robust clipping bounds. PMTD employs a progressive distillation strategy with both full-precision and multiple high-bit (INT) teachers to enhance low-bit models' capacity and quality. Extensive experiments demonstrate that our method outperforms existing approaches, achieving state-of-the-art performance across multiple tasks and benchmarks. The code will be made publicly available.

## 1 Introduction

Multi-frame video enhancement tasks aim to enhance the spatial and temporal resolution and quality of video sequences by exploiting temporal information from multiple frames. Among these, Video Frame Interpolation (VFI) [1, 20, 25, 39, 44, 46, 50, 53, 78–80, 86, 89], Video Super-Resolution (VSR) [3–6, 9, 22, 26], and Spatio-Temporal Video Super-Resolution (STVSR) [12, 17, 38, 66, 71, 72, 74, 87] are the most representative video enhancement methods. They are widely employed as post-processing and pre-processing techniques in social media platforms, gaming environments, and various video perception and generation tasks [4, 12]. Recent Transformer-based approaches for video enhancement exploit attention mechanisms to capture temporal dependencies across multiple frames, enabling substantial improvements in visual quality and structural fidelity. However, their high computational and memory demands remain a major obstacle for real-world deployment. Therefore, numerous

---

*\* Renjing Pei, Yang Wang and Yang Cao are the corresponding authors. † ZhanFeng Feng, Long Peng, and Xin Di contributed equally to this work. ‡ Long Peng is the project leader.

39th Conference on Neural Information Processing Systems (NeurIPS 2025).

studies have proposed various model quantization methods to compress the bit-width of weights and activations from 32 bits (FP32) to 8, 4, or 2 bits [19, 28, 40–42, 56, 57, 76]. This is a crucial step in practical deployment, reducing memory consumption and inference latency. For example, PAMS [28], a classic method in Post-Training Quantization (PTQ), introduces trainable scale parameters to dynamically learn the maximum value of the quantization range. Liu *et al.* propose 2DQuant [40], a dual-stage method for image super-resolution which designs a differentiated search strategy and uses knowledge distillation [16] to guide the learning of the quantization range. However, to the best of our knowledge, exploring model quantization in video enhancement remains largely uncharted. Directly applying existing quantization methods can lead to significant issues such as performance degradation and loss of fine details. Through observations and statistical experiments, we attribute these problems to two key limitations: (a) Video enhancement models need to aggregate texture and motion information from multiple frames, leading to inter-frame differential perception of information, manifesting as differentiated activation value distributions across frames as shown in Figure 2(a). However, traditional quantization methods fails to allocate inconsistent representational capacity to different frames, resulting in discrepancies in dynamic range across frames, as shown in Figure 2(b). This results in suboptimal utilization of sub-pixel spatial details, thereby limiting reconstruction performance. (b) Quantization inevitably reduces the representational capacity of the video model. Traditional methods overlook the capacity differences between the teacher (FP32) and student models (2bit, 4bit), relying solely on full-precision teachers for distillation. This makes it challenging for the student to learn high-quality mappings given its limited capacity, resulting in difficulty when directly quantizing a high-precision network into a low-precision one, as shown in Figure 1(a). To address these limitations, we propose a novel quantization framework: Progressive Multi-Frame Quantization for multi-frame Video Enhancement, called PMQ-VE. Specifically, PMQ-VE introduces a coarse-to-fine two-stage process, which includes Backtracking-based Multi-Frame Quantization (BMFQ) and Progressive Multi-Teacher Distillation (PMTD).

**Coarse Stage: Backtracking-based Multi-Frame Quantization.** Existing methods [40, 64] typically initialize quantization bounds using the global minimum and maximum values and symmetrically shrink them inward, ignoring inter-frame variations and the asymmetric nature of distributions. To address this, as illustrated in Figure 2(c), BMFQ assigns frame-specific clipping bounds to better match the heterogeneous activation statistics across video frames. BMFQ employs a percentile-based initialization to suppress outliers and performs a backtracking-based search with pruning and backtracking to search the bounds efficiently. This strategy enables accurate, adaptive quantization with negligible overhead,

**Fine Stage: Progressive Multi-Teacher Distillation.** In the fine stage, we introduce a Progressive Multi-Teacher Distillation framework to restore the model's representational capacity under low-bit quantization. Specifically, a full-precision teacher provides fine-grained feature supervision, while an intermediate 8/4-bit teacher offers quantization-aware guidance, helping the 4/2-bit student model learn stable and informative representations under low-bit constraints, bridging the gap between the quantized and full-precision models.

Extensive experiments on three representative video enhancement tasks—STVSR, VSR, and VFI—demonstrate that our method achieves state-of-the-art performance across multiple benchmarks for various tasks. Our method consistently outperforms existing methods, achieving the best performance on PSNR, SSIM across various bit-width settings, as shown in Figure 1(b-c). The contributions of this paper can be summarized as follows:

- To the best of our knowledge, we are the first to explore model quantization in multi-frame video enhancement tasks. We propose PMQ-VE, a novel per-frame coarse-to-fine quantization framework for multi-frame video enhancement models.

- In the coarse stage, we propose BMFQ to search quantization bounds via iterative backtracking with pruning, achieving efficient initialization. In the fine stage, we propose PMTD to leverage the knowledge of multi-level teachers to help a low-bit student model, enhancing its mapping quality and performance.

- Extensive experiments on three video enhancement tasks (STVSR, VSR, and VFI) demonstrate that our method achieves state-of-the-art results across various benchmarks under different low-bit quantization settings, highlighting the superiority and practicability of our approach.

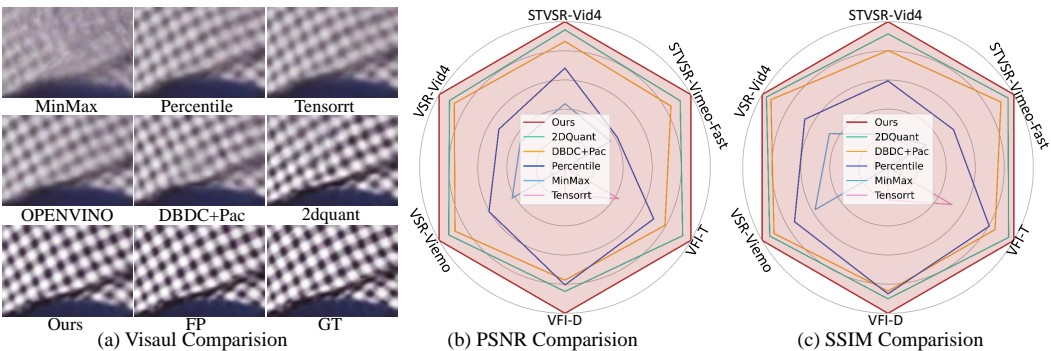

Figure 1: (a) Qualitative comparison of reconstructed frames using different quantization methods. Quantitative comparison of PSNR(b) and SSIM(c) improvements across three video enhancement tasks (STVSR, VFI, VSR). Our method consistently outperforms existing quantization approaches in both visual quality and quantitative metrics.

# 2 Related work

## 2.1 Video Enhancement

Video enhancement aims to exploit sub-pixel information from contextual frames to improve the quality and resolution of videos, which primarily includes video frame interpolation, video super-resolution, and spatio-temporal video super-resolution.

**Video Frame Interpolation (VFI)** targets generating the intermediate frames between given consecutive inputs. Early CNN-based methods [1, 20, 25, 44, 89] mainly rely on optical flow estimation or direct frame synthesis, but often suffer from limited receptive fields and poor handling of large motion. Therefore, Transformer-based approaches [39, 53, 86] have been proposed to model long-range dependencies, significantly improving the quality and detail of generated video.

**Video Super-Resolution (VSR)** aims to reconstruct high-resolution (HR) video from low-resolution (LR) inputs. Early VSR methods primarily used explicit optical flow alignment [3, 5, 63], dynamic filtering [26], deformable convolutions [67], and temporal attention mechanisms [30, 73]. With the increasing prominence of the Transformer's powerful representation capabilities, numerous Transformer-based VSR methods have been proposed, achieving progressive success. For example, PSRT [62] leverages a multi-frame self-attention mechanism to jointly process features from the current input frame and the propagated features. MIA [88] further boosts performance by leveraging masked intra-frame and inter-frame attention blocks to better use of previously enhanced features.

**Spatio-Temporal Video Super-Resolution (STVSR)** aims to simultaneously enhance spatial and temporal resolution, combining VSR and VFI, and presents greater challenges. Among the most representative real-time Transformer-based models is RSTT [12], which achieves state-of-the-art performance by constructing feature dictionaries from different levels of encoders and repeatedly querying them during the decoding stage.

Although powerful transformer-based models have demonstrated superiority in enhancing spatial resolution and perceptual quality, their high computational cost hinders practical deployment. This paper is the first to propose an efficient model compression method specifically for video enhancement to facilitate its deployment.

## 2.2 Model Quantization

Model quantization aims to reduce the model's bit-width, from the Float 32-bit used in training to int 8, 4, or 2-bit for deployment, significantly reducing computational and memory costs and is widely applied in various fields such as LLM and VLM, etc. Quantization is divided into post-training quantization (PTQ) and Quantization-Aware Training (QAT). QAT, requiring simultaneous training and quantization, demands significant resources and data. PTQ, applied after pre-training, is more efficient and thus receives greater research focus. Early PTQ methods focused on minimizing quantization error using efficient calibration techniques [34, 76]. Recent approaches, such as AdaRound [70] and BRECQ [33], refine weight quantization by minimizing layer-wise output discrepancies. Additionally, robustness-oriented methods like NoisyQuant [42], OASQ [47], and ERQ [85] enhance PTQ performance by mitigating quantization noise, suppressing outliers, or optimizing error-aware objectives.

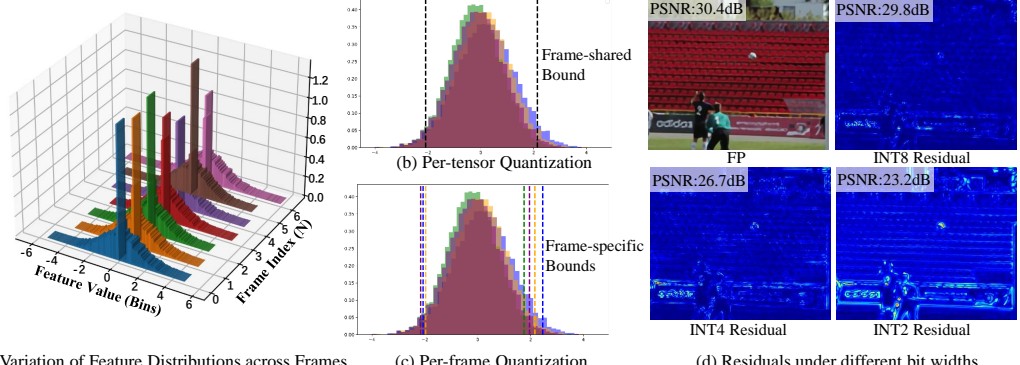

(a) Variation of Feature Distributions across Frames    (b) Per-tensor Quantization    (c) Per-frame Quantization    (d) Residuals under different bit widths

Figure 2: Finding and Motivation. (a) In multi-frame video enhancement, activation distributions vary significantly across frames. Traditional per-tensor quantization (b) fails to dynamically adjust quantization bounds for these variations, but our method (c) achieves this dynamic adjustment. (d) We calculated PSNR and residual maps for FP, INT8, INT4, and INT2 with respect to GT. The significant gap between low-bit (INT2/4) and full-precision (FP) model suggests that low-bit struggles to learn directly from FP. This inspired us to use multiple teacher models for supervision.

However, existing work mainly addresses high-level vision/language tasks and is often unsuitable for pixel-level image/video enhancement, which is sensitive to quantization errors due to its reliance on fine-grained features. Recent research has thus started exploring quantization for pixel-level image enhancement and super-resolution [7, 8, 37, 61, 82]. For example, DBDC+Pac [64] introduces a PTQ framework for image super-resolution, combining calibration techniques with knowledge distillation from a full-precision teacher model. Similarly, 2DQuant [40] targets SwinIR [35] by proposing a one-sided search algorithm to quantize sensitive activations, such as post-softmax and post-GELU [18] layers. These methods often overlook inter-frame differences in multi-frame video enhancement, limiting detail representation and resulting in blurred images.

## 3 Methodology

### 3.1 Problem Formulation

Model quantization aims to learn appropriate clipping ranges [40, 64, 76] for each tensor (e.g., weights or activations) in order to minimize the discrepancy between the outputs of the full-precision model and the quantized model. Following previous work [40, 41, 64], we use fake quantization [24] to simulate the quantization process. Given a pre-learned clipping range $[lb, ub]$ for a tensor $x$, the quantization and dequantization process is defined as follows:

$$x_{\text{clip}} = \text{clamp}(x, lb, ub) = \min(\max(x, lb), ub), \tag{1}$$

$$x_{\text{int}} = \text{round}\left(\frac{x_{\text{clip}} - lb}{\Delta}\right), \quad \Delta = \frac{ub - lb}{2^N - 1}, \quad \hat{x} = x_{\text{int}} \cdot \Delta + lb, \tag{2}$$

Where $x_{\text{clip}}$ is the clipped tensor, $x_{\text{int}} \in \{0, 1, \dots, 2^N - 1\}$ is the quantized integer, $\Delta$ is the quantization step size, and $\hat{x}$ is the dequantized approximation of the original value. Linear and MatMul layers are the most computationally intensive components in Transformer-based architectures. We follow existing work [40, 42] to focus quantization on these modules. Since the quantization function is non-differentiable due to rounding, we also adopt the Straight-Through Estimator (STE) [10] during training to approximate gradients and enable end-to-end optimization under quantized settings. More details are in the **Appendix A**.

### 3.2 Observations and Motivation

**Observation 1: Inability to allocate varying representational capacity across frames.** Previous studies have thoroughly examined the statistical properties of activations in Transformer-based architectures, uncovering long-tailed distributions and a mix of symmetric and asymmetric behaviors across layers [34, 40, 49, 76, 85]. However, these analyses are largely confined to single-frame scenarios. In the context of quantizing multi-frame video enhancement networks, it is essential

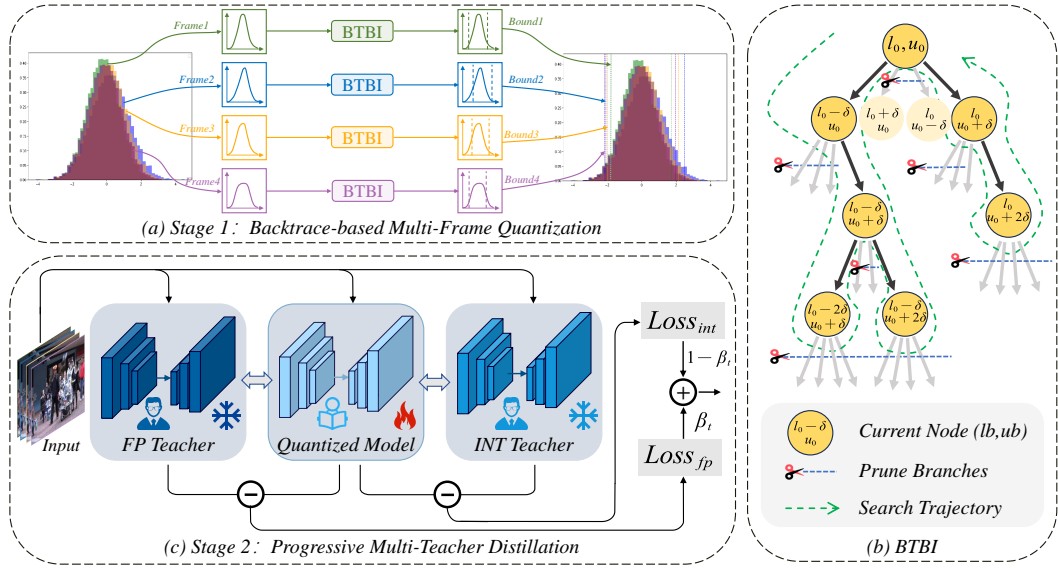

Figure 3: The overall framework of our proposed method.

to preserve the capability of the full-precision network to effectively integrate inter-frame texture information and motion cues—a challenge that vanilla methods fail to address.

Our analysis reveals that multi-frame networks perceive and process each frame in the input tensor differently. As shown in Figure 2(a), we collect per-frame activation statistics and observe significant disparities in activation distributions across frames. In particular, the value ranges (i.e., the minimum and maximum activation values) vary considerably, indicating that the network allocates representational capacity unevenly across frames. These differences are influenced by the model's frame-dependent attention dynamics. In particular, the network assigns different attention weights to each frame, resulting in varied activation distributions. Consequently, applying existing Transformer quantization methods [34, 40, 49, 76, 85], which typically assume a unified activation distribution, can be suboptimal for multi-frame models. Such methods overlook inter-frame variation, leading to inefficient quantization and increased error, ultimately degrading the quality of the output.

**Observation 2: Over-reliance on full-precision teachers limits low-bit model learning.** Low-bit quantization inevitably reduces the representational capacity of models, which brings significant challenges to multi-frame video enhancement tasks. Under low-bit quantization, both activation values and network precision degrade noticeably, making it difficult to maintain the clear motion trajectories and rich texture details required for these tasks. As shown in Figure 2(c), directly quantizing the network to 4-bit/2-bit and applying it without adaptation leads to severe artifacts and motion blur. The main reason for this degradation lies in the large quantization errors introduced when networks are directly quantized to low-bit precision. Furthermore, traditional methods often overlook the capacity gap between the teacher model (FP32) and the student model (e.g., 4-bit or 2-bit), relying solely on full-precision teachers for knowledge distillation [16, 40, 64]. This makes it difficult for the student model to learn high-quality mappings within its limited capacity, further increasing the challenge of achieving satisfactory performance under low-bit constraints.

### 3.3 Proposed Method

Based on the above observations, we propose a two-stage quantization framework. The coarse stage uses Backtracking-based Multi-Frame Quantization (BMFQ) to initialize asymmetric bounds efficiently. The fine stage applies Progressive Multi-Teacher Distillation (PMTD) to refine bounds.

#### 3.3.1 Backtracking-based Multi-Frame Quantization

Motivated by Observation 1, we adopt a per-frame quantization strategy to handle frame-wise variations in activation distributions and representational capacity. Given a multi-frame activation tensor $X \in \mathbb{R}^{N \times C \times H \times W}$, where $N$ denotes the number of frames, we independently search the clipping bounds for each frame $X_i = X[i, :, :, :]$, yielding frame-specific clipping parameters $(lb_i, ub_i)$ for $i = 1, \ldots, N$. This strategy enables the quantizer to adapt to frame-specific activation

statistics, thereby improving quantization accuracy.

To robustly estimate the clipping bounds during the coarse stage, we formulate the selection of $(lb_i, ub_i)$ as a constrained optimization problem that minimizes quantization-induced distortion over a percentile-based search space $S_i$ derived from the empirical distribution of $X_i$:

$$(lb_i^*, ub_i^*) = \underset{(lb, ub) \in S_i}{\arg\min} \ \mathbb{E}_{x \sim X_i} \left[ (x - Q_{lb, ub}(x))^2 \right], \tag{3}$$

Here, $Q_{lb, ub}(\cdot)$ denotes a uniform quantizer [43] with clipping range $[lb, ub]$. To mitigate the influence of outliers, we constrain the search space $S_i$ using percentiles: $lb \in [p_{0.1}(X_i), p_{10}(X_i)]$, $ub \in [p_{90}(X_i), p_{99.9}(X_i)]$, where $p_k(X)$ denotes the $k$-th percentile of the tensor $X$. To efficiently solve the optimization problem in Eq. (3), we introduce a Backtracking-based Bound Initialization (BTBI) algorithm. Starting from an initial estimate derived from the percentiles of $X_i$, the algorithm recursively explores candidate bounds by adjusting $lb_i$ and $ub_i$ within the search space $S_i$. At each step, it evaluates the quantization error and updates the optimal bounds if a better configuration is found. To avoid redundant searches, previously visited bounds are skipped. The algorithm backtracks to explore alternative adjustments when no further improvement is achieved, terminating when all candidates are evaluated or a convergence threshold is met. In contrast to traditional methods that uniformly shrink bounds [40] or adjust them sequentially [64], BTBI is less sensitive to outliers and explores a richer set of candidate configurations. By combining frame-wise adaptation with recursive backtracking search, BTBI robustly converges to optimal clipping parameters for each frame. To enhance understanding, we provide a detailed algorithm of our BTBI in Algorithm 1.

---

**Algorithm 1** Backtracking-based Bound Initialization (BTBI) pipeline

**Input:** $X$, step sizes $\Delta L$, $\Delta U$, threshold $\varepsilon$
**Output:** Optimal bounds $lb^*$, $ub^*$
$visited \leftarrow \emptyset$ , $error_{\min} \leftarrow \infty$
**Function** Backtrack($lb, ub$):

> **if** $(lb, ub) \in visited$ *or out-of-range*
> **then**
>   ⌊ **return**
> $visited \leftarrow visited \cup \{(lb, ub)\}$
> $X_q \leftarrow$ Quantize $X$ using $(lb, ub)$
> $err \leftarrow \|X - X_q\|_2$
> **if** $err > error_{\min} + \varepsilon$ **then**
>   ⌊ **return**
> **if** $err < error_{\min}$ **then**
>   | $error_{\min} \leftarrow err$
>   | $lb^* \leftarrow lb$
>   ⌊ $ub^* \leftarrow ub$
> **foreach** $(\delta_l, \delta_u) \in \{\pm\Delta L, \pm\Delta U\}$ **do**
>   ⌊ Backtrack($lb + \delta_l, ub + \delta_u$)

Backtrack($lb_0, ub_0$)
**return** $lb^*, ub^*$

---

### 3.3.2 Progressive Multi-Teacher Distillation

As revealed by Observation 2, training accurate quantized models under extremely low-bit settings (e.g., 4-bit or 2-bit) remains challenging due to limited capacity and optimization instability. To address this, we propose Progressive Multi-Teacher Distillation (PMTD), a hierarchical distillation framework that leverages both high-bit and full-precision teachers to facilitate low-bit training. Instead of directly distilling knowledge from a full-precision (FP) teacher to a low-bit student, which often suffers from large representational gaps, PMTD introduces intermediate-bit teacher models (e.g., 8-bit) between the full-precision (FP) teacher and the low-bit student. These intermediate teachers serve as quantization-aware approximations of the FP model, providing smoother supervision and easing the optimization process. This hierarchical approach effectively bridges the representational gap between student and teacher models, ensuring more stable training dynamics and improved performance under extreme quantization constraints. Specifically, to train a 4-bit quantized model, PMTD first uses the full-precision model as a teacher to train an 8-bit model. When training the 4-bit model, both the full-precision network and the 8-bit network are used as teacher models, as illustrated in Figure 3(c). The distillation process is formally defined by the following loss function:

$$\mathcal{L}_{\text{PMTD}} = (\mathcal{L}_{\text{INT}} + \alpha(t) \cdot \mathcal{L}_{\text{FP}})/(1 + \alpha(t)), \tag{4}$$

where $\mathcal{L}_{\text{INT}}$ denotes the total loss from intermediate-bit teachers (e.g., 8-bit), and $\mathcal{L}_{\text{FP}}$ represents the loss from the full-precision teacher. The balancing coefficient $\alpha(t)$ linearly increases over time and is defined as $\alpha(t) = \min\left(1, \frac{t}{T_{\text{warmup}}}\right)$, where $T_{\text{warmup}}$ is a hyperparameter controlling the warm-up duration. Each teacher-specific loss consists of two components: an output-level reconstruction loss

Table 1: Quantitative comparison of different methods on four STVSR benchmarks. The best and the second best results are in **bold** and bold.

| Method | Bit | Vid4 | | Vimeo-Fast | | Vimeo-Medium | | Vimeo-Slow | |
|---|---|---|---|---|---|---|---|---|---|
| | | PSNR↑ | SSIM↑ | PSNR↑ | SSIM↑ | PSNR↑ | SSIM↑ | PSNR↑ | SSIM↑ |
| RSTT-S [12] | 32/32 | 26.29 | 0.7941 | 36.58 | 0.9381 | 35.43 | 0.9358 | 33.30 | 0.9123 |
| Trilinear | 32/32 | 22.90 | 0.5883 | 29.45 | 0.8019 | 29.16 | 0.8351 | 28.21 | 0.8091 |
| OpenVINO [15] | 2/2 | 18.24 | 0.3151 | 23.39 | 0.6103 | 23.60 | 0.6227 | 23.87 | 0.6242 |
| TensorRT [65] | 2/2 | 20.31 | 0.5118 | 23.41 | 0.6106 | 23.61 | 0.6236 | 23.88 | 0.6283 |
| SNPE [21] | 2/2 | 15.22 | 0.2378 | 23.40 | 0.6106 | 23.61 | 0.6241 | 23.88 | 0.6281 |
| Percentile [29] | 2/2 | 12.67 | 0.1349 | 15.27 | 0.2274 | 14.80 | 0.2165 | 14.67 | 0.2156 |
| MinMax [23] | 2/2 | 10.34 | 0.0138 | 10.52 | 0.0266 | 10.48 | 0.0289 | 10.45 | 0.0303 |
| NoisyQuant [42] | 2/2 | 12.06 | 0.1028 | 12.50 | 0.1669 | 11.81 | 0.1465 | 11.49 | 0.1508 |
| DBDC+Pac [64] | 2/2 | 22.64 | 0.5695 | 28.94 | 0.8254 | 28.87 | 0.8214 | 27.86 | 0.7905 |
| 2DQuant [40] | 2/2 | 22.91 | 0.5883 | 29.38 | 0.8315 | 29.14 | 0.8330 | 28.18 | 0.8086 |
| **Ours** | 2/2 | **23.48** | **0.6252** | **30.33** | **0.8424** | **30.19** | **0.8523** | **29.14** | **0.8316** |
| OpenVINO [15] | 4/4 | 18.84 | 0.4591 | 21.82 | 0.6372 | 21.63 | 0.6315 | 18.84 | 0.4591 |
| TensorRT [65] | 4/4 | 18.63 | 0.4578 | 21.74 | 0.6019 | 21.70 | 0.6324 | 18.63 | 0.4578 |
| SNPE [21] | 4/4 | 17.84 | 0.3977 | 21.64 | 0.6018 | 21.56 | 0.6301 | 18.76 | 0.4584 |
| Percentile [29] | 4/4 | 23.26 | 0.6314 | 27.12 | 0.7664 | 27.16 | 0.7709 | 26.58 | 0.7531 |
| MinMax [23] | 4/4 | 21.60 | 0.5242 | 26.41 | 0.6990 | 25.94 | 0.7059 | 25.44 | 0.6957 |
| NoisyQuant [42] | 4/4 | 24.26 | 0.6905 | 31.22 | 0.8719 | 30.64 | 0.8705 | 29.61 | 0.8462 |
| DBDC+Pac [64] | 4/4 | 24.50 | 0.6923 | 32.64 | 0.8857 | 32.06 | 0.8866 | 30.64 | 0.8643 |
| 2DQuant [40] | 4/4 | 25.04 | 0.7256 | 33.59 | 0.9035 | 32.83 | 0.9009 | 31.21 | 0.8766 |
| **Ours** | 4/4 | **25.42** | **0.7501** | **34.69** | **0.9181** | **33.74** | **0.9150** | **31.94** | **0.8903** |

and an intermediate feature-matching loss:

$$\mathcal{L}_{\text{INT}} = \sum_{k=1}^{K} \left( \mathcal{L}_{\text{rec}}^{(k)} + \lambda \cdot \mathcal{L}_{\text{feat}}^{(k)} \right), \tag{5}$$

$$\mathcal{L}_{\text{FP}} = \mathcal{L}_{\text{rec}}^{\text{FP}} + \lambda \cdot \mathcal{L}_{\text{feat}}^{\text{FP}}, \tag{6}$$

where $K$ is the number of intermediate-bit teachers (e.g., $K = 2$ when using 4-bit and 8-bit teachers), $\mathcal{L}_{\text{rec}}$ is the $\ell_1$ loss [36] between the student and teacher outputs, and $\mathcal{L}_{\text{feat}}$ is the mean squared error (MSE) [2] between selected intermediate feature representations. The balancing coefficient $\lambda$ is set to 5 to emphasize the importance of internal consistency. By gradually transitioning supervision from intermediate-bit to full-precision teachers, PMTD effectively reduces the training difficulty of low-bit models, mitigates quantization errors, and offers a more stable optimization path. This hierarchical approach ensures high-quality quantized outputs, even under extreme quantization constraints.

## 4 Experiment and Analysis

### 4.1 Experiment Setting

**Datasets and backbone.** We evaluate our method on three representative video enhancement tasks: Space-Time Video Super-Resolution (STVSR), Video Super-Resolution (VSR), and Video Frame Interpolation (VFI). For each task, we select state-of-the-art and popular methods as backbones: RSTT [12] for STVSR, MIA [88] for VSR, and EMA-VFI [86] for VFI. The Vimeo-90K [74] dataset is used for training across all tasks, with Vid4 [38] and the Vimeo-90K test set serving as evaluation benchmarks. More details of data preparation and setting are provided in the supplementary material.
**Evaluation metrics.** We use PSNR and SSIM [68] as evaluation metrics, computed on the luminance (Y) channel of the YCbCr color space. To further evaluate perception-oriented metrics, LPIPS [81] and NIQE [48] are used to assess the perceptual quality of videos.
**Implementation details.** We adopt the Adam optimizer [27] with an initial learning rate of $2 \times 10^{-4}$

Table 2: Quantitative comparison of different methods on two VFI EMA-VFI variants ([T] and [D]), evaluated on the Vimeo90K benchmark under 4-bit.

| Method | Bit | EMA-VFI [T] [78] | | | | EMA-VFI [D] [86] | | | |
|---|---|---|---|---|---|---|---|---|---|
| | | PSNR↑ | SSIM↑ | LPIPS↓ | NIQE↓ | PSNR↑ | SSIM↑ | LPIPS↓ | NIQE↓ |
| Baseline | 32/32 | 29.41 | 0.9279 | 0.086 | 6.736 | 30.29 | 0.9418 | 0.078 | 6.545 |
| OpenVINO [15] | 4/4 | 26.03 | 0.8703 | 0.222 | 8.022 | 25.38 | 0.8579 | 0.257 | 8.2784 |
| TensorRT [65] | 4/4 | 25.33 | 0.8551 | 0.268 | 8.582 | 25.21 | 0.8537 | 0.269 | 8.4035 |
| SNPE [21] | 4/4 | 25.49 | 0.8581 | 0.259 | 8.500 | 25.83 | 0.8683 | 0.236 | 8.0399 |
| Percentile [29] | 4/4 | 26.82 | 0.8919 | 0.185 | 7.765 | 28.54 | 0.9198 | 0.132 | 7.0667 |
| MinMax [23] | 4/4 | 23.03 | 0.7918 | 0.389 | 9.475 | 24.19 | 0.8309 | 0.313 | 8.5153 |
| DBDC+Pac[64] | 4/4 | 27.30 | 0.8976 | 0.171 | 7.545 | 28.36 | 0.9179 | 0.134 | 7.1221 |
| 2DQuant [40] | 4/4 | _28.06_ | _0.9110_ | _0.152_ | _7.494_ | _28.78_ | _0.9233_ | _0.120_ | _6.9884_ |
| Ours | 4/4 | **28.41** | **0.9162** | **0.136** | **7.361** | **29.59** | **0.9335** | **0.101** | **6.7881** |

Table 3: Quantitative comparison of different methods on two VSR benchmarks under 4-bit.

| Benchmark | Metric | Baseline: MIA [88] | TensorRT [65] | SNPE [21] | Percentile [29] | MinMax [23] | DBDC +Pac [64] | 2DQuant [40] | Ours |
|---|---|---|---|---|---|---|---|---|---|
| Vimeo90K | PSNR↑ | 38.32 | 31.81 | 32.42 | 35.15 | 34.13 | 36.64 | _36.92_ | **37.34** |
| | SSIM↑ | 0.9532 | 0.8612 | 0.8805 | 0.9262 | 0.9119 | 0.9404 | _0.9434_ | **0.9487** |
| Vid4 | PSNR↑ | 28.20 | 24.48 | 24.22 | 26.14 | 25.60 | 27.26 | _27.38_ | **27.64** |
| | SSIM↑ | 0.8507 | 0.6758 | 0.6877 | 0.7805 | 0.7494 | 0.8230 | _0.8287_ | **0.8341** |

and apply Cosine Annealing [45] over 20,000 iterations. The batch size is set to 8 and 2 per GPU during the initialization and distillation-based fine-tuning phases, respectively. Random cropping, rotations, and flipping are applied to enhance training robustness. All experiments are implemented in Python with PyTorch [54] and conducted on 8 NVIDIA V100 GPUs.

## 4.2 Quantitative Results

Table 1 presents quantitative comparisons of various methods under 2/2, 4/4, bit-width across four STVSR benchmarks. Traditional quantization approaches, such as OpenVINO [15] and TensorRT [65], face challenges in pixel-level video enhancement, resulting in model performance scores of only 18.24dB and 20.31dB on Vid4 at 2-bit quantization. Although DBDC+Pac [64] and 2DQuant [40] are tailored for low-level vision tasks with enhanced sharpness awareness, which somewhat mitigates the performance drop due to quantization, they still lag behind our proposed method. This is primarily due to their limitations in managing multi-frame distribution differences and detail enhancement. Our method achieves the best performance across all scenarios and benchmarks, notably surpassing existing methods by nearly 1 dB on the Vimeo benchmark, underscoring the effectiveness of our approach. In a similar manner, our method demonstrates superior performance across all benchmarks and bit-width configurations for both Video Super-Resolution (VSR) and Video Frame Interpolation (VFI) tasks. As detailed in Tables 2 and 3, our approach exemplifies remarkable generalization capabilities, consistently outperforming existing methods in all benchmarks and bits. More results on additional bit-width settings and benchmarks can be found in **Appendix E**.

## 4.3 Qualitative Results

To verify the visual quality, we present the visual comparisons of different PTQ methods applied to STVSR, VSR and VFI tasks under 4-bit quantization, as shown in Figure 4. Traditional methods such as MinMax [23] and Percentile [29] exhibit noticeable artifacts, while other methods like DBDC+Pac [64] and 2dquant [40] suffer from detail blurring issues, particularly in complex scenes. However, our proposed method consistently produces sharper edges and more faithful textures that are visually closer to the full-precision outputs. In more challenging cases, such as the Vimeo-Fast dataset where motion and fine details coexist, our proposed method better preserves structural information

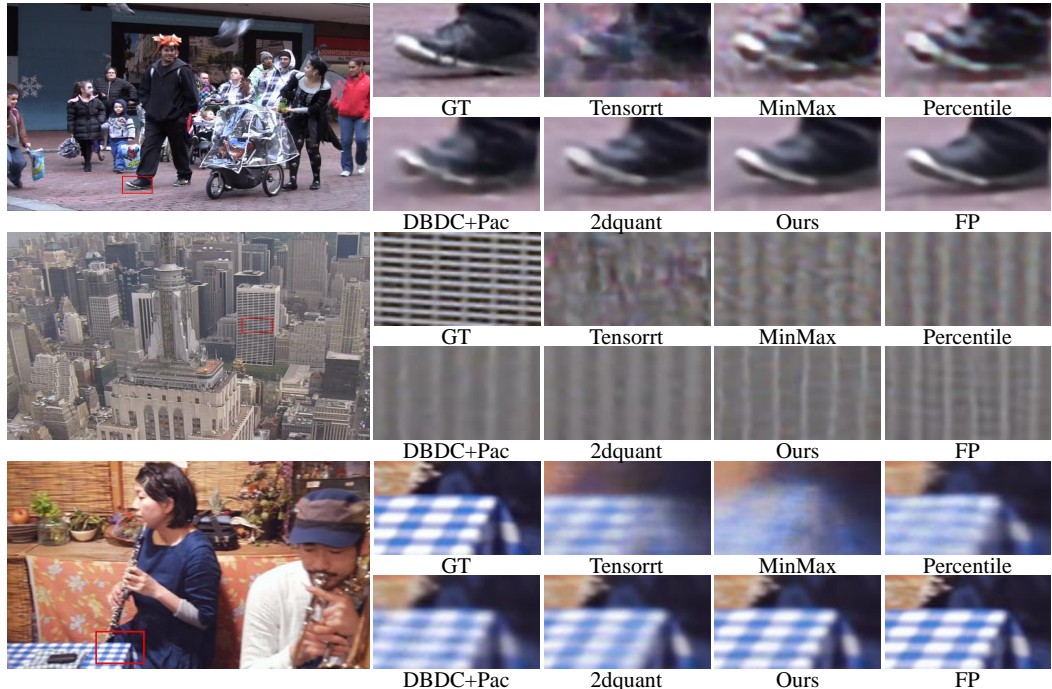

Figure 4: Visual comparisons under 4-bit quantization for three video enhancement tasks: from top to bottom are STVSR, VSR, and VFI tasks. More results are provided in Appendix F.

and avoids common artifacts. This highlights the visual superiority of our approach. More visual effects and comparisons with user studies will be presented in the **Appendix F**.

## 4.4 Ablation Studies

To validate the effectiveness of the proposed core idea, we design several ablation experiments to explore the multi-teacher distillation strategy the frame-wise quantization strategy. Specifically, we conduct experiments on STVSR in a 2-bit compression setting, removing these core modules one by one, with results shown in Table 6. It is seen that the baseline without any core ideas achieves only 12.67dB. By introducing the frame-wise quantization strategy, the network perceives differences between frames, improving performance. Furthermore, the BMFQ helps the network adaptively learn clipping ranges for each frame, boosting

Table 4: Ablation studies.

| Per-Frame Quantization | BMFQ | PMTD | PSNR↑ |
|:---:|:---:|:---:|:---:|
| ✗ | ✗ | ✗ | 12.67 |
| ✓ | ✗ | ✗ | 19.64 |
| ✓ | ✓ | ✗ | 27.56 |
| ✓ | ✓ | ✓ | 30.33 |

model performance to 27.56dB. Finally, with the introduction of the multi-teacher distillation strategy, the low-bit network learns prior knowledge from different teachers, further improving model performance despite limited capacity. This validates the effectiveness of the proposed core modules. **More ablation studies are presented in the Appendix H.**

## 5 Conclusion

We introduced a novel coarse-to-fine PMQ-VE, addressing key challenges in quantizing multi-frame video enhancement models. BMFQ is proposed to establish robust quantization bounds through a percentile-based initialization and backtracking search, ensuring efficient quantization across frames. PMTD enhances the quality of low-bit models by utilizing a progressive distillation strategy with both full-precision and quantized teachers, bridging the gap between high-precision and low-bit models. Experiments on STVSR, VSR, and VFI tasks show that our PMQ-VE achieves state-of-the-art performance and visually pleasing results.

Limitation and Future Work: Although our PMQ-VE has achieved promising results on Transformer-based video enhancement methods, more diffusion-based Transformer (DiT) methods can be tested. In the future, we plan to extend our method to more video enhancement tasks and models to facilitate the deployment of video models in the community.

## Acknowledgments

This work was supported by the Natural Science Foundation of China under Grants 62225207,62436008 and 62206262. The AI-driven experiments, simulations and model training were performed on the robotic AI-Scientist platform of Chinese Academy of Sciences.

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

# Appendix / Supplemental material

## A   More Details about Quantization Scheme

This section provides further elaboration on the quantization process introduced in the problem formulation. We detail the quantization strategy employed for Transformer-based architectures and describe how gradient approximation is performed using the Straight-Through Estimator (STE), enabling end-to-end training under quantized constraints.

### A.1   Quantization Strategy for Transformer Blocks

Following prior work [40, 42], we apply quantization to the most computationally intensive components of Transformer models, including all linear transformations and batched matrix multiplications. To simulate quantization during training, we adopt a fake quantization approach, wherein both weights and activations are quantized using floating-point operations that emulate integer arithmetic. Figure 5 illustrates the quantization workflow within a typical Transformer block, covering the QKV attention mechanism, Multi-Head Self-Attention (MSA), and the feed-forward network (MLP). All Q-Linear layers operate on quantized weights and activations [13, 14, 69, 75, 77, 83, 84], which are processed through dedicated quantizers prior to arithmetic operations. For simplicity, dropout layers in the attention and projection modules are omitted in the illustration.

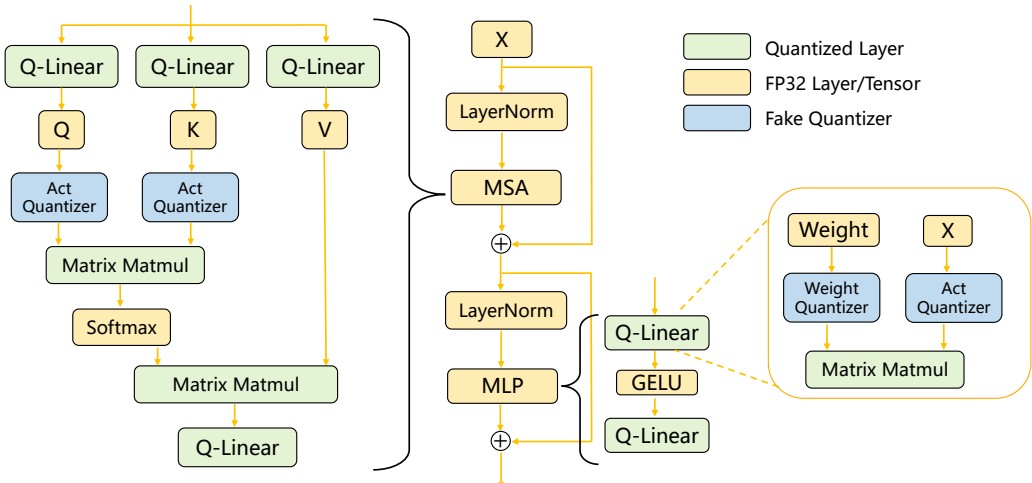

Figure 5: Quantization scheme for Transformer blocks. Green boxes denote quantized layers, yellow boxes indicate FP32 operations and tensors, and blue boxes represent fake quantizers. `Q-Linear` indicates a quantized linear transformation. Each matrix multiplication is performed on quantized weights and activations.

### A.2   Gradient Approximation with STE

However, both the Clip and Round operations are non-differentiable, which impedes the use of gradient-based optimization methods [31, 32, 51, 52]. To overcome this, we adopt the Straight-Through Estimator (STE), where the gradients are approximated during backpropagation. Specifically, for the Round operation, the gradient is approximated as:

$$\frac{\partial \text{Round}(x)}{\partial x} = 1. \tag{7}$$

For the Clip operation, the partial derivatives are approximated as:

$$\frac{\partial \text{Clip}(x, l, u)}{\partial x} = \begin{cases} 1, & \text{if} \quad l \leq x \leq u, \\ 0, & \text{otherwise}, \end{cases} \tag{8}$$

$$\frac{\partial \text{Clip}(x, l, u)}{\partial l} = \begin{cases} 1, & \text{if} \quad x < l, \\ 0, & \text{otherwise}, \end{cases} \qquad \frac{\partial \text{Clip}(x, l, u)}{\partial u} = \begin{cases} 1, & \text{if} \quad x > u, \\ 0, & \text{otherwise}. \end{cases} \tag{9}$$

By leveraging these approximations, PTQ frameworks can fine-tune the clipping parameters $\{l_w, u_w\}$ for weights and $\{l_a, u_a\}$ for activations, enabling better adaptation to low-bit constraints without altering the original floating-point model parameters.

## A.3 Distinguishing Our PTQ Approach from QAT

To prevent potential ambiguity, we explicitly define our methodology as **Post-Training Quantization (PTQ)**. A core principle of our approach is that throughout the entire quantization process, the weights of the pre-trained full-precision model are kept completely frozen and are never updated [11, 55, 57–60]. The optimization is exclusively aimed at determining the optimal quantization parameters, specifically the clipping bounds for both weights and activations.

A key aspect of our method is the use of fake quantization and the Straight-Through Estimator (STE) during the optimization of these clipping bounds. It is crucial to understand that these techniques are employed solely as a mechanism to enable gradient-based optimization of the clipping bounds through backpropagation. This strategy of optimizing quantization parameters while keeping weights fixed is a common and advanced technique in the PTQ literature, enabling more accurate parameter learning without the significant computational overhead of full model retraining [40, 76].

## A.4 Clarification on Multi-Frame Quantization

We wish to clarify the term "multi-frame quantization" to accurately reflect its implementation and scope. The term does not imply that we store a unique set of quantization parameters for every single frame of an entire video sequence. Such an approach would be impractical, leading to prohibitive storage costs and failing to generalize to videos of arbitrary length.

Instead, our definition of multi-frame quantization is tailored to the operational architecture of modern video enhancement models. These models, whether based on sliding-window or recurrent structures, process a fixed number ($N$) of frames at each inference step. For instance, a sliding-window model takes a local window of $N$ frames as input to produce an enhanced central frame.

Consequently, our "per-frame quantization" refers to the practice of learning and storing $N$ distinct sets of clipping bounds—one for each of the $N$ frame slots within this fixed-size processing window. This fine-grained approach captures local temporal dynamics more effectively, leading to superior quantization performance.

# B   More Details of Datasets and Training Settings.

In this work, we use the Vimeo-90K septuplet dataset [74] as the primary training dataset across all three tasks. The Vimeo-90K dataset contains 64,612 training clips and 7,824 testing clips, with each clip consisting of seven frames at a spatial resolution of $448 \times 256$. For evaluation, in addition to the Vimeo-90K test set, we also adopt the Vid4 dataset [38], a classical benchmark containing four video sequences (calendar, city, foliage, and walk). During the PTQ training phase, we adopt a knowledge distillation strategy where the supervision relies on teacher model predictions rather than HR references. However, the usage of the Vimeo-90K dataset slightly varies across tasks due to the differences in task requirements and experimental protocols, as detailed below.

For the STVSR task, we use the Vimeo-90K dataset [74] for training. each video clip in Vimeo-90K is treated as high-resolution (HR) and high-frame-rate (HFR) references. The low-resolution (LR) and low-frame-rate (LFR) inputs are generated by selecting odd-index frames from the HR sequences and downsampling them by a factor of 4 using bicubic interpolation. Following standard practices in previous STVSR studies [12, 66, 72], we evaluate the model on both the Vid4 dataset and the Vimeo-90K test set. The Vimeo-90K test set is further divided into Vimeo-Slow, Vimeo-Medium, and Vimeo-Fast subsets based on motion magnitude. To avoid infinite PSNR values, we exclude six sequences from the Vimeo-Medium subset and three from the Vimeo-Slow subset, as suggested in [66].

For the VSR task, we follow the experimental settings of prior works [5, 6, 62]. The Vimeo-90K dataset is used for training, with its seven-frame clips serving as training samples. For evaluation, we adopt both the Vimeo-90K test set (Vimeo90K-T) and the Vid4 dataset to benchmark model performance.

For the VFI task, we train our models on the Vimeo-90K septuplet dataset by following the experimental settings introduced in [88]. Two training paradigms are employed: (1) the **T** setting, which follows the traditional arbitrary time indexing paradigm where temporal interpolation is

performed at uniformly sampled time steps, and (2) the **D** setting, which adopts a distance-based indexing strategy, prioritizing frames closer to the input during interpolation. In line with [88], we use the first and last frames of each sequence as inputs to predict the intermediate five frames.

## C Detailed Structure of Backbones.

### C.1 Backbone for STVSR Task: RSTT

The Real-time Spatial Temporal Transformer (RSTT) [12] is a cascaded UNet-style architecture for joint temporal interpolation and spatial super-resolution. It consists of four encoder stages $E_k$ and four corresponding decoder stages $D_k$ $(k = 0, 1, 2, 3)$.

Given four consecutive low-resolution, low-frame-rate input frames $\{I_{2t-1}^L, I_{2t+1}^L, I_{2t+3}^L, I_{2t+5}^L\}$, the encoders extract multi-scale spatio-temporal features, which are stored as dictionaries. A query builder generates seven queries $\{Q_1, Q_2, \ldots, Q_7\}$ used to synthesize the target high-resolution, high-frame-rate frames via the decoder stages.

**Encoder Block.** Each encoder block consists of two sub-blocks: a Window-based Multi-head Self Attention (W-MSA) block and a Shifted Window-based Multi-head Self Attention (SW-MSA) block. Each sub-block is wrapped with Layer Normalization (LN), a multi-layer perceptron (MLP), and residual connections.

The computation in each encoder block is:

$$X = \text{W-MSA}(\text{LN}(X)) + X, \tag{10}$$
$$X = \text{MLP}(\text{LN}(X)) + X, \tag{11}$$
$$X = \text{SW-MSA}(\text{LN}(X)) + X, \tag{12}$$
$$X = \text{MLP}(\text{LN}(X)) + X. \tag{13}$$

For a windowed feature $X \in \mathbb{R}^{M^2 \times C}$, attention is computed as:

$$\text{Attention}(Q, K, V) = \text{Softmax}\left(\frac{QK^\top}{\sqrt{d}}\right) V, \quad Q = XW^Q, \ K = XW^K, \ V = XW^V. \tag{14}$$

**Decoder Block.** Each decoder block shares the same structure as the encoder block, except that self-attention is replaced by cross-attention. Specifically, W-MSA and SW-MSA are replaced by: Window-based Multi-head Cross Attention (W-MCA) and Shifted Window-based Multi-head Cross Attention (SW-MCA) The cross-attention is computed as:

$$\text{CrossAttention}(Q, K, V) = \text{Softmax}\left(\frac{QK^\top}{\sqrt{d}}\right) V. \tag{15}$$

In this case, the query $Q$ comes from the decoder input (or previous output), while the key and value $(K, V)$ are obtained from the encoder dictionaries at the corresponding level.

### C.2 Backbone for VSR Task: MIA

The overall structure of MIA-VSR [88] follows the bi-directional second-order grid propagation framework as in BasicVSR++ [6]. The network consists of three main parts: feature extraction, feature propagation, and feature reconstruction. The **feature extraction** module uses convolution to extract shallow features from the input low-resolution (LR) frames. The **feature reconstruction** module adopts a pixel-shuffle layer to upsample and reconstruct the high-resolution (HR) frames. The core of the network lies in the middle: the **feature propagation module**, where the MIA blocks are applied to recurrently enhance features across frames. Each feature propagation module (FPM) consists of $N$ cascaded Masked Inter&Intra-frame Attention Blocks (IIABs). For video frame $t$, the $m$-th FPM receives the current input $\boldsymbol{X}_m^t$ and the enhanced features from previous frames $\boldsymbol{X}_{m+1}^{t-1}, \boldsymbol{X}_{m+1}^{t-2}$ to produce the updated feature $\boldsymbol{X}_{m+1}^t$:

$$\boldsymbol{X}_{m+1}^t = \text{FPM}(\boldsymbol{X}_m^t, \boldsymbol{X}_{m+1}^{t-1}, \boldsymbol{X}_{m+1}^{t-2}). \tag{16}$$

Each block inside FPM performs the following update:

$$\boldsymbol{X}_{m,n+1}^t = \text{IIAB}(\boldsymbol{X}_{m,n}^t, \boldsymbol{X}_{m+1}^{t-1}, \boldsymbol{X}_{m+1}^{t-2}), \tag{17}$$

where $n = 0, \ldots, N - 1$ and $\boldsymbol{X}_{m,0}^t = \boldsymbol{X}_m^t$. The attention mechanism is used inside each IIAB block. Instead of computing joint self-attention over concatenated features, MIA decouples attention into intra-frame and inter-frame branches. Specifically: The **Query** $\boldsymbol{Q}_{m,n}^t$ is generated only from the current feature $\boldsymbol{X}_{m,n}^t$. The **Keys** and **Values** contain both intra-frame tokens (from $\boldsymbol{X}_{m,n}^t$) and inter-frame tokens (from $\boldsymbol{X}_{m+1}^{t-1}, \boldsymbol{X}_{m+1}^{t-2}$). The QKV formulation in the IIAB block is as follows:

$$
\begin{aligned}
\boldsymbol{Q}_{m,n}^t &= \boldsymbol{X}_{m,n}^t W^Q, \\
\boldsymbol{K}_{m,n}^t &= \left[ \boldsymbol{X}_{m,n}^t W^K ; \left[ \boldsymbol{X}_{m+1}^{t-1} ; \boldsymbol{X}_{m+1}^{t-2} \right] W^K \right], \\
\boldsymbol{V}_{m,n}^t &= \left[ \boldsymbol{X}_{m,n}^t W^V ; \left[ \boldsymbol{X}_{m+1}^{t-1} ; \boldsymbol{X}_{m+1}^{t-2} \right] W^V \right],
\end{aligned}
\tag{18}
$$

where $W^Q, W^K, W^V$ are learnable projection matrices.

The attention output is computed as:

$$
\texttt{Attn} = \text{Softmax} \left( \frac{\boldsymbol{Q}_{m,n}^t {\boldsymbol{K}_{m,n}^t}^\top}{\sqrt{d}} + B \right) \boldsymbol{V}_{m,n}^t,
\tag{19}
$$

where $B$ is a learnable relative positional bias.

The output is then passed through a feedforward network (FFN) with LayerNorm and residual connections as in standard Transformer blocks.

### C.3   Backbone for VFI Task: EMA-VFI

The overall architecture of EMA-VFI [78] consists of three main stages: (1) a CNN-based multi-scale feature extractor, (2) a Transformer-based motion-appearance feature extractor, and (3) a lightweight flow estimation module. The `FlowEstimation` module takes motion features $\boldsymbol{M}_{0 \to 1}$, $\boldsymbol{M}_{1 \to 0}$ and appearance features $\boldsymbol{A}_0, \boldsymbol{A}_1$ from the Transformer, and outputs refined flows $\boldsymbol{F}_{t \to 0}$, $\boldsymbol{F}_{t \to 1}$ along with a blending mask $\boldsymbol{O}_t$ to synthesize the intermediate frame.

**Transformer Block.** Each Transformer block is designed to jointly extract appearance and motion features using inter-frame attention. The block follows the standard Transformer layout with two residual sub-layers: an attention module and a feed-forward network (FFN):

$$
\boldsymbol{X}' = \boldsymbol{X} + \texttt{InterFrameAttention}(\texttt{LN}(\boldsymbol{X})),
\tag{20}
$$

$$
\boldsymbol{X}^{\text{out}} = \boldsymbol{X}' + \texttt{FFN}(\texttt{LN}(\boldsymbol{X}')),
\tag{21}
$$

where $\boldsymbol{X}$ is the input token feature derived from either $I_0$ or $I_1$.

**Inter-frame Attention.** The attention is computed between a patch in $I_0$ and its spatial neighbors in $I_1$, formulated as:

$$
\boldsymbol{Q}_{i,j} = \boldsymbol{A}_0^{i,j} W^Q,
\tag{22}
$$

$$
\boldsymbol{K}_{n(i,j)} = \boldsymbol{A}_1^{n(i,j)} W^K, \quad \boldsymbol{V}_{n(i,j)} = \boldsymbol{A}_1^{n(i,j)} W^V,
\tag{23}
$$

$$
\boldsymbol{S}_{i,j}^{0 \to 1} = \text{Softmax} \left( \frac{\boldsymbol{Q}_{i,j} \cdot \boldsymbol{K}_{n(i,j)}^\top}{\sqrt{d}} \right),
\tag{24}
$$

where $n(i, j)$ denotes the spatial neighborhood around position $(i, j)$.

**Feed-forward Network (FFN).** The FFN module incorporates depth-wise convolution to enhance spatial modeling:

$$
\texttt{FFN}(\boldsymbol{X}) = \texttt{DWConv}(\texttt{GELU}(\boldsymbol{X} W_1)) W_2,
\tag{25}
$$

where `DWConv` denotes a depth-wise convolution operator.

The Transformer blocks are stacked to extract hierarchical motion and appearance features from multi-scale CNN features, providing rich spatiotemporal representations for accurate intermediate frame synthesis.

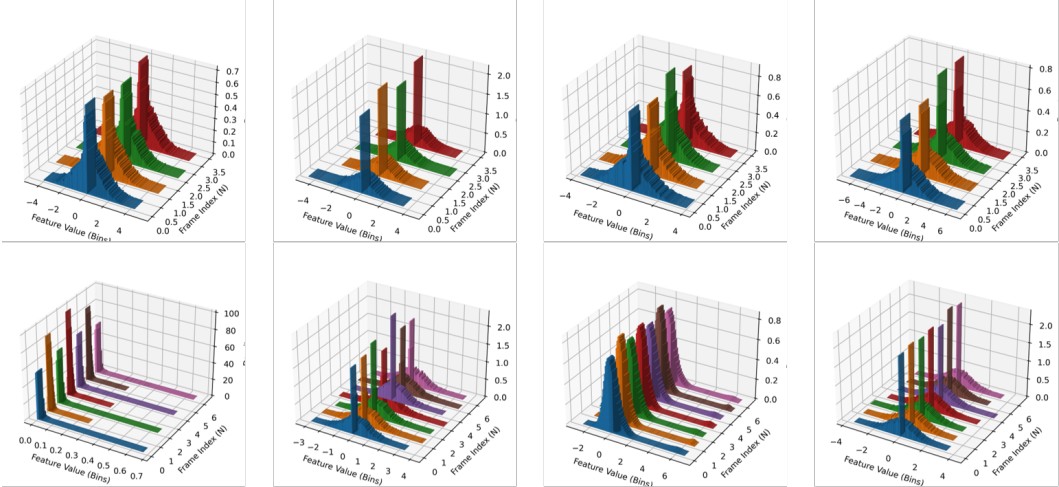

Figure 6: Per-frame activation statistics from Transformer layers. Each curve corresponds to a different frame in the input tensor.

Table 5: Quantitative comparison of different methods on four STVSR benchmarks under 6-bit quantization. The best and second best results are in **bold** and bold.

| Method | Bit | Vid4 | | Vimeo-Fast | | Vimeo-Medium | | Vimeo-Slow | |
| --- | --- | --- | --- | --- | --- | --- | --- | --- | --- |
| | | PSNR↑ | SSIM↑ | PSNR↑ | SSIM↑ | PSNR↑ | SSIM↑ | PSNR↑ | SSIM↑ |
| OpenVINO [15] | 6/6 | 25.78 | 0.7692 | 35.49 | 0.9279 | 34.60 | 0.9259 | 32.67 | 0.8985 |
| TensorRT [65] | 6/6 | 25.71 | 0.7690 | 35.47 | 0.9275 | 34.60 | 0.9254 | 32.60 | 0.8981 |
| SNPE [21] | 6/6 | 25.92 | 0.7735 | 35.86 | 0.9315 | 34.71 | 0.9272 | 32.74 | 0.9035 |
| Percentile [29] | 6/6 | 25.88 | 0.7732 | 35.66 | 0.9298 | 34.64 | 0.9269 | 32.73 | 0.9035 |
| MinMax [23] | 6/6 | 25.96 | 0.7742 | 35.92 | 0.9315 | 34.69 | 0.9267 | 32.69 | 0.9021 |
| NoisyQuant [42] | 6/6 | 25.92 | 0.7741 | 35.81 | 0.9317 | 34.75 | 0.9281 | 32.70 | 0.9021 |
| DBDC+Pac [64] | 6/6 | 26.05 | 0.7837 | 36.10 | 0.9338 | 34.94 | 0.9303 | 32.92 | 0.9068 |
| 2DQuant [40] | 6/6 | 26.04 | 0.7819 | 36.12 | 0.9339 | 34.95 | 0.9302 | 32.93 | 0.9068 |
| **Ours** | 6/6 | **26.10** | **0.7850** | **36.21** | **0.9345** | **35.04** | **0.9311** | **33.00** | **0.9076** |

## D  More Visualizations about Inter-frame Differences.

To further support the observation of frame-wise representational disparity discussed in Section 3.2, we present additional visualizations of activation distributions and attention patterns across frames in a multi-frame video enhancement model.

Figure 6 shows the per-frame activation value ranges collected from intermediate Transformer layers. We observe significant variation in both the minimum and maximum activation values across frames, indicating frame-dependent distribution shifts. The figure illustrates that the model dynamically attends to frames with varying intensity, further confirming that representational capacity is not uniformly allocated. These observations justify the need for frame-aware quantization strategies that account for inter-frame variation in activation statistics and attention dynamics.

## E  More Quantitative Results under additional bit-width

Table 5 presents the quantitative comparison of various quantization methods on four STVSR benchmarks under 6-bit quantization. As shown, our method consistently outperforms all existing approaches across all datasets in terms of both PSNR and SSIM.

## F  More Visual Comparisons

To further validate the visual effectiveness of our method, we present additional qualitative comparisons under 4-bit post-training quantization in Figure 7. The figure includes challenging examples from three representative video enhancement tasks: Space-Time Video Super-Resolution (STVSR), Video Super-Resolution (VSR), and Video Frame Interpolation (VFI).

As shown in the figure, traditional methods such as MinMax [23] and Percentile [29] often result in oversmoothed regions or noticeable artifacts. More advanced baselines like DBDC+Pac [64] and 2dquant [40] provide better structure but still suffer from detail loss, especially in textured or high-motion areas. In contrast, our method produces sharper edges and more faithful textures that are visually closer to the full-precision (FP) results.

These results further support the claim that our quantization approach better preserves perceptual quality across a wide range of scenes and motion patterns.

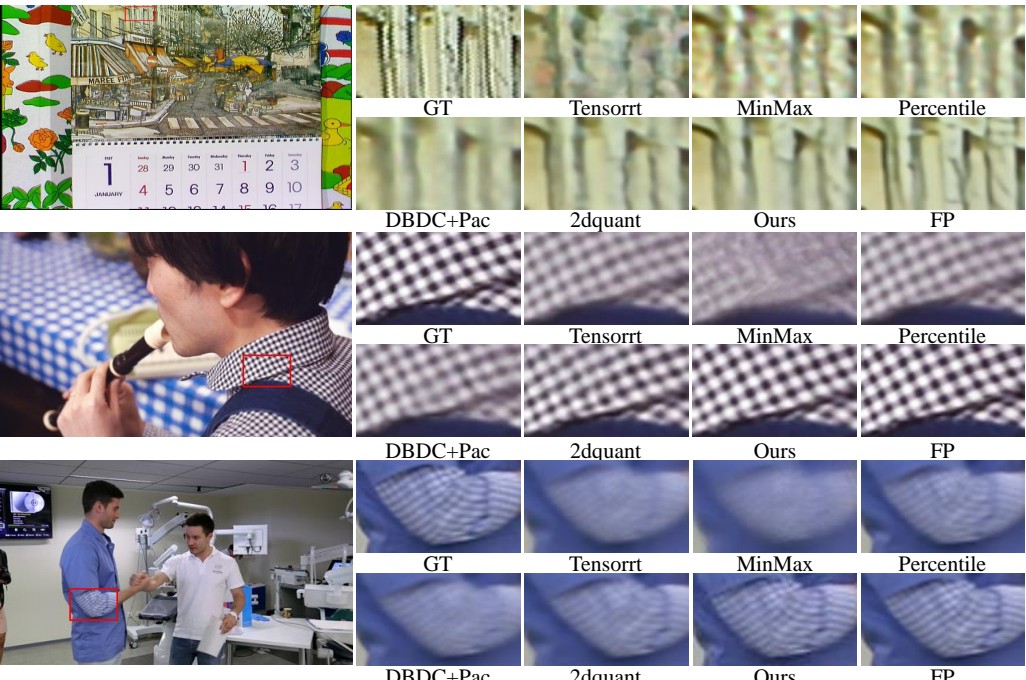

Figure 7: Additional visual comparisons under 4-bit quantization for three video enhancement tasks: STVSR, VSR, and VFI. Our method consistently produces outputs closer to the ground truth (GT) and full-precision (FP) results.

Table 6: Quantitative results under the 2-bit setting. PSNR and SSIM values are reported for multiple benchmark datasets. The best and the second best results are in **bold** and bold.

| Method | Bit | Set5 (x2) | | Set14 (x2) | | B100 (x2) | | Urban100 (x2) | | Manga109 (x2) | |
|---|---|---|---|---|---|---|---|---|---|---|---|
| | | PSNR↑ | SSIM↑ | PSNR↑ | SSIM↑ | PSNR↑ | SSIM↑ | PSNR↑ | SSIM↑ | PSNR↑ | SSIM↑ |
| SwinIR-light [35] | 32 | 38.15 | 0.9611 | 33.86 | 0.9206 | 32.31 | 0.9012 | 32.76 | 0.9340 | 39.11 | 0.9781 |
| Bicubic | 32 | 32.25 | 0.9118 | 29.25 | 0.8406 | 28.68 | 0.8104 | 25.96 | 0.8088 | 29.17 | 0.9128 |
| MinMax [23] | 2 | 33.88 | 0.8951 | 30.05 | 0.8428 | 29.99 | 0.7232 | 26.30 | 0.7378 | 32.57 | 0.9446 |
| Percentile [29] | 2 | 34.35 | 0.9015 | 30.55 | 0.8466 | 30.45 | 0.7276 | 26.73 | 0.7464 | 33.10 | 0.9502 |
| DBDC+Pac [64] | 2 | 34.55 | 0.9052 | 30.65 | 0.8487 | 30.55 | 0.7291 | 26.85 | 0.7493 | 33.25 | 0.9529 |
| DOBI [40] | 2 | 35.25 | 0.9361 | 31.72 | 0.8917 | 30.62 | 0.8699 | 28.52 | 0.8727 | 33.65 | 0.9529 |
| 2DQuant [40] | 2 | 36.00 | 0.9497 | 31.98 | 0.9012 | 30.91 | 0.8810 | 28.62 | 0.8819 | 34.40 | 0.9602 |
| **BTBI+PMTD** | 2 | **36.07** | **0.9501** | **32.04** | **0.9014** | **31.00** | **0.8821** | **28.68** | **0.8905** | **34.45** | **0.9615** |

# G   User Stduy

To evaluate the visual quality of the image super-resolution results, we conducted a user study involving 15 participants. Fifteen images were randomly selected from the test datasets, and the participants were asked to rate the quality of each processed image on a scale from 0 (poor quality) to 10 (excellent quality). The aggregated results, as illustrated in the figure, indicate that existing methods struggle to fully restore image quality, resulting in lower user satisfaction. In contrast, our method achieves the highest average score of 8.1, significantly outperforming other methods and demonstrating superior capabilities in visual performance and generalization.

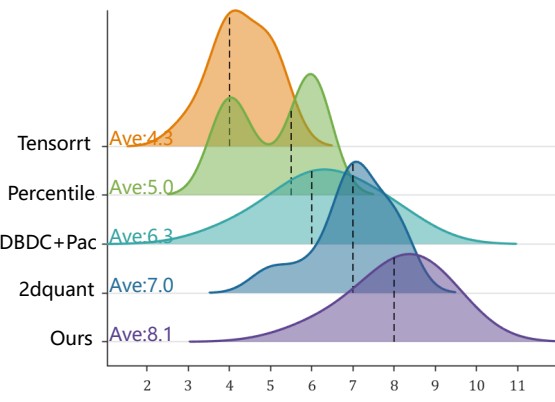

Figure 8: User study on image visual quality.

# H   More Ablation Studys

In this section, we provide additional ablation studies to verify the effectiveness of our proposed method. Specifically, we replace the core components of 2DQuant [40] with our approach. That is, we replace DOBI with BTBI and DQC with our multi-teacher distillation network. Following the experimental setup in 2DQuant, we conduct experiments under the 2-bit setting. The results demonstrate that our method achieves significant improvements over the baseline, as shown in Table 6.

