# OpenReview forum: "PMQ-VE: Progressive Multi-Frame Quantization for Video Enhancement"
_NeurIPS.cc/2025/Conference — NeurIPS 2025 poster_

### Official Review · Reviewer_B6R2 · 2025-06-28

**Clarity:** 3
**Significance:** 3
**Originality:** 3
**Rating:** 5
**Confidence:** 3

**Summary:**

The authors propose Backtracking-based Multi-Frame Quantization, a simple but effective framework that addresses the limitations of traditional global quantization in multi-frame video enhancement. Besides, the authors propose PMTD, a method that leverages knowledge from multi-level teacher models to guide a low-bit student model, thereby improving its mapping quality and overall performance. The structure of the paper is clear, the motivation is explicit, and the experimental results demonstrate the effectiveness of the two innovations proposed by the authors.

**Questions:**

1. Instead of searching for the optimal upper and lower bounds for each frame, searching every few frames (because the spatial information of adjacent frames of the video is extremely close) can we further reduce the computational complexity of the method?
2. What is the impact of varying the number of intermediate-bit teachers (e.g., using 6-bit alongside 8-bit) in the PMTD strategy on low-bit model performance?
3. How does the percentile-based initialization in BMFQ specifically suppress outliers, and what percentile ranges are experimentally validated as optimal?

**Ethical Concerns:**

["NO or VERY MINOR ethics concerns only"]

**Final Justification:**

The author has excellently demonstrated through experiments the trade-off between different quantization levels and the number of quantization frames and the computational efficiency. After considering the opinions of other reviewers, I have decided to increase my score.

**Limitations:**

Not applicable.

**Paper Formatting Concerns:**

Not applicable.

**Quality:**

3

**Strengths And Weaknesses:**

Strengths:
1. The paper proposes a two-stage PMQ-VE framework and progressive multi-teacher distillation (PMTD) to address inter-frame distribution differences and capacity gaps in multi-frame video enhancement.
2. The percentile-based initialization and backtracking search efficiently optimize frame-specific bounds with negligible overhead.
3. Extensive experiments across STVSR, VSR, and VFI demonstrate their effectiveness.

Weaknesses:
1. The study lacks analysis of computational trade-offs across different quantization levels (e.g., 2-bit vs. 4-bit) for practical edge deployment scenarios.

---

> ### Author Rebuttal · Authors · 2025-07-31
>
> We appreciate the reviewer’s recognition of the novelty and effectiveness of our frame-wise quantization strategy and the progressive multi-teacher distillation framework. We also appreciate the positive feedback regarding the clarity of our paper and the strong experimental results demonstrated across STVSR, VSR, and VFI tasks.
>
> > **Q1:** Computational Trade-offs across Different Quantization Levels
>
> **A1:** We quantitatively analyze the trade-offs between bit-width, visual quality, model compression, and inference speed. As shown in Table 1, 4-bit quantization strikes a good balance,  achieving  \~3×  compression, \~1.5× speedup, and retaining over 96% of the full-precision PSNR. In contrast, 2-bit models offer the highest efficiency, but at the cost of a significant quality drop (\~2.8dB PSNR loss).
>
> #### Table 1: Trade-offs Between Bit-width, Efficiency, and Quality (Vid4)
> | Bit-width | PSNR (Vid4) | SSIM (Vid4) | Compression ratio | Speedup ratio |
> | --------- | ----------- | ----------- | ----------------- | ------------- |
> | 32-bit    | 26.29       | 0.7941      | 1x                | 1x            |
> | 8-bit     | 26.24       | 0.7939      | 2.39x             | 1.22x         |
> | 4-bit     | 25.42       | 0.7501      | 3.11x             | 1.53x         |
> | 2-bit     | 23.48       | 0.6252      | 3.67x             | 2.01x         |
>
> > **Q2:** Reducing Complexity via Frame Grouping for Bound Search
>
> **A2:** We conduct experiments by modifying the granularity of quantization bound search from per-frame to per-group, where one group contains multiple consecutive frames. Using our two-stage framework, we tested different group sizes (1, 2, 3, 4). In preliminary experiments, we observed that grouping frames and sharing quantization bounds within short temporal windows (e.g., every 2–3 frames) can reduce the computational overhead of BTBI with negligible impact on accuracy. However, for fast motion or complex scenes, where inter-frame differences are more significant, per-frame quantization still offers better robustness. The results (on STVSR task, 2-bit setting) are summarized in Table 2:
>
> #### Table 2: Impact of Frame Grouping on BTBI Efficiency and Quantization Performance
> | Group Size    | BTBI Cost Time | BTBI itertion | PSNR (Vid4) | PSNR (Vimeo-Fast) | PSNR (Vimeo-Medium) | PSNR (Vimeo-Slow) |
> | ------------- | -------------- | ------------- | ----------- | ----------------- | ------------------- | ----------------- |
> | 1 (per-frame) | 70.89s          | 410.52        | 23.48       | 30.33             | 30.19               | 29.14             |
> | 2             | 53.56s          | 369.32        | 23.36       | 30.11             | 30.13               | 29.10             |
> | 3             | 42.91s          | 301.54        | 23.11       | 30.02             | 30.00               | 28.99             |
> | 4             | 30.01s          | 241.98        | 22.99       | 29.48             | 29.71               | 28.78             |
>
> > **Q3:** Effect of Adding Intermediate-Bit Teachers in PMTD
>
> **A3:** We extended our PMTD framework by introducing a 6-bit teacher, trained via distillation from FP32 and 8-bit models. We then evaluated 4-bit and 2-bit students with and without the 6-bit teacher. As shown in Table 3, adding a 6-bit teacher generally improves performance, suggesting smoother supervision and enhanced knowledge transfer.
>
> However, the gains are not always substantial, and come at the cost of increased memory and training overhead. This demonstrates that our PMTD framework enables effective trade-offs between accuracy and resource consumption.
>
> #### Table 3: Impact of Intermediate-Bit Teachers on Student Performance and Training Overhead
> | Student | Teachers             | Peak Mem (GB) | PSNR (Vid4) | PSNR (Vimeo-Fast) | PSNR (Vimeo-Medium) | PSNR (Vimeo-Slow) |
> | ------- | -------------------- | ------------- | ----------- | ----------------- | ------------------- | ----------------- |
> | 4-bit   | FP32 + 8-bit         | 12.3          | 25.42       | 34.69             | 33.74               | 31.94             |
> | 4-bit   | FP32 + 8-bit + 6-bit | 16.3          | 25.46       | 34.69             | 33.75               | 31.94             |
> | 2-bit   | FP32 + 8-bit + 4-bit | 16.2          | 23.48       | 30.33             | 30.19               | 29.14             |
> | 2-bit   | FP32 + 8 + 6 + 4-bit | 20.5          | 23.48       | 30.36             | 30.21               | 29.13             |
>
> > **Q4:** Effectiveness of Percentile-Based Initialization in Suppressing Outliers and Optimal Range Selection
>
> **A4:** In BMFQ, we initialize the clipping bounds using fixed percentile ranges ([0.1–10] for lower bounds, [90–99.9] for upper bounds) instead of min-max. This helps suppress outliers by explicitly excluding the extreme tails of activation distributions, which are often noisy and unrepresentative. This design is motivated by empirical observation: after BTBI and PMTD optimization, most learned clipping points lie within this percentile range. We further validated this choice through an ablation study (Table 4), where our default setting consistently outperforms both wider and narrower ranges.
>
> In summary, percentile-based initialization provides a robust starting point that avoids distortion from outliers while preserving meaningful dynamic range.
> #### Table 4: Impact of Percentile Initialization Ranges on Quantization Performance (4-bit)
> | Variant          | Range (lower/upper)      | PSNR (Vid4) | PSNR (Vimeo-Fast) | PSNR (Vimeo-Medium) | PSNR (Vimeo-Slow) |
> | ---------------- | ------------------------ | ----------- | ----------------- | ------------------- | ----------------- |
> | Default          | [0.1, 10] / [90, 99.9]   | **24.15**   | **32.24**         | **30.77**           | **30.66**         |
> | Wider Range ↑    | [0.01, 15] / [85, 99.99] | 23.99       | 32.01             | 29.86               | 30.24             |
> | Narrower Range ↓ | [0.1, 1] / [99, 99.9]    | 23.91       | 32.10             | 29.84               | 30.22             |

---

> > ### Comment · Reviewer_B6R2 · 2025-08-06
> >
> > Thank you to the author for your detailed reply, which effectively answered my questions. After considering the opinions of other reviewers, I have decided to increase my score.

---

### Official Review · Reviewer_7rvP · 2025-07-01

**Clarity:** 1
**Significance:** 3
**Originality:** 3
**Rating:** 4
**Confidence:** 4

**Summary:**

This paper proposes PMQ-VE, a novel two-stage quantization framework for multi-frame video enhancement tasks. PMQ-VE introduces Backtracking-based Multi-Frame Quantization to search frame-specific clipping bounds via percentile initialization and backtracking search, and Progressive Multi-Teacher Distillation to leverage full-precision and intermediate-bit teachers for knowledge transfer. Experiments show PMQ-VE outperforms SOTA methods across benchmarks under low-bit quantization.

**Questions:**

1. The main problem is: Is BMFQ performed during the training phase (i.e., QAT, to learn frame-specific bounds for the training set) or during inference for each test frame (i.e., PTQ)? I suppose BMFQ might be a QAT strategy, but the authors have not explicitly pointed this out. If it is training-phase only, it seems that you are overfitting lb and ub parameters for each specific frame in the training set, then how do the learned bounds generalize to unseen test frames with different distributions? If it's done at test phase, it means you are quantizing a full-precision model during the test phase. Since this full-precision model, which can lead to better image quality than the quantized model, can be used for inference during testing, then what's the point and meaning of doing quantization? Additionally, if it's done at test phase, how much is the computational cost? Please clarify the workflow.

2. The paper claims BTBI has "negligible overhead," but please provide quantitative comparisons (e.g., training iterations, GPU memory usage, inference latency) between PMQ-VE and baseline methods to validate this. If overhead exists, how does it impact edge deployment scenarios where efficiency is critical?

3. In PMTD, how are the intermediate-bit teachers trained? Are they quantized from the full-precision model using the same BMFQ strategy, or via other methods?

I'm not sure if my understanding is entirely correct so far. If the authors can address my concerns well, I might increase my evaluation score; otherwise, I will still tend to reject the manuscript

**Ethical Concerns:**

["NO or VERY MINOR ethics concerns only"]

**Final Justification:**

The authors have cleared up my misunderstanding and addressed my concerns during the rebuttal. The proposed approach now seems technically novel and sound. I suggest that the authors improve the clarity of the writing and incorporate some key issues we discussed into the final version.

**Limitations:**

yes

**Quality:**

2

**Strengths And Weaknesses:**

Strengths: 1. The novel BMFQ and PMTD strategies address the challenges in video enhancement quantization. 2. The authors conducted comprehensive experiments and ablation studies on different tasks and benchmarks, which demonstrates quantitative and qualitative improvements.

Weakness
1.The main problem is the clarity of BMFQ implementation. The motivation of BMFQ is also not well introduced. The paper lacks explanation on whether BMFQ is applied during training or testing, and how frame-specific bounds are generalized to unseen test frames (if BMFQ is applied to the training set). This ambiguity causes lots of confusion about the practical deployment workflow (e.g., if per-frame boundary search is required for each test input, it may introduce considerable inference computational cost; if the per-frame boundary search is done for each training input, the parameters would not be suitable for the test frames).
2. Besides, the backtracking search in BMFQ is described as "negligible overhead," but the paper does not provide concrete data (e.g., training/inference time increase, memory usage) to support this claim, which is critical for evaluating practical efficiency.

---

> ### Author Rebuttal · Authors · 2025-07-31
>
> We appreciate the reviewer’s recognition of our novel quantization strategies and the effectiveness of our empirical validation across multiple video enhancement tasks. In response to the reviewer’s questions, we provide the following clarifications:
>
> > **Q1:** What's the point and meaning of doing quantization? Is BMFQ a PTQ or QAT method?
>
> **A1:**  Quantization reduces model size and inference cost by converting high-precision computations into low-precision ones, enabling efficient deployment without significant accuracy loss. Our approach is **Post-Training Quantization (PTQ)**. Specifically, we apply BMFQ and PMTD using unlabeled data, without updating the model weights. The purpose is to learn quantization parameters for both weights and activations. Consistent with other PTQ methods, our approach does not run during inference, ensuring no additional overhead at deployment time.
>
> To clarify:
> - QAT updates both weights and quantization parameters during training.
> - PTQ keeps weights fixed and only learns clipping bounds.
>
> QAT provides higher accuracy by simulating quantization during training, but it requires retraining and more resources. PTQ is simpler and faster to apply, making it suitable for efficient deployment without modifying the original training pipeline — that’s why we choose PTQ.
>
> > **Q2:** How frame-specific bounds are generalized to unseen test frames?
>
> **A2:** We'd like to clarify that our approach follows the standard practice of all PTQ methods, where quantization parameters (e.g., clipping bounds) are learned once offline from a calibration set and then frozen for inference, This is a widely adopted paradigm in PTQ literature [1, 2, 3]
>
> Regarding generalization to unseen frames, we note that activation ranges are generally stable across samples, thanks in large part to the use of normalization layers (e.g., LayerNorm) in Transformer-based video enhancement models. While the distribution shape of activations may vary depending on content or motion, the value range—which is the key factor for quantization—remains relatively consistent across different samples.
>
> Since these ranges are primarily determined by the  model architecture and its pretrained parameters—rather than by the raw input pixels or the specific temporal position of a frame—they tend to remain consistent across different sequences. This enables the learned clipping bounds to generalize well to unseen data.
>
> > **Q3:** The motivation of BMFQ is not well introduced
>
> **A3:**  We would like to clarify that this has been discussed in detail in Lines 152–168 of the main paper. The motivation for proposing Backtracking-based Multi-Frame Quantization (BMFQ) stems from two key challenges in quantizing multi-frame video enhancement models:
> 1. Per-frame activation variation:
>    As shown in Figure 2-a of our paper, we collect per-frame activation statistics and observe significant disparities in activation distributions across frames. Traditional quantization methods — using shared clipping bounds across frames — fail to capture this variation, leading to suboptimal quantization and degraded reconstruction quality. BMFQ addresses this by introducing frame-specific bounds, allowing for better adaptation to heterogeneous activation statistics.
> 2. Robust and efficient bound initialization:
>    Our backtracking strategy refines the initialization by recursively searching within a constrained percentile range. This approach avoids exhaustive scanning and ensures high-quality quantization with negligible training overhead.
>
> Additionally, as demonstrated in our ablation study, directly applying PMTD from a naive min-max initialization leads to suboptimal results. In contrast, using BMFQ provides a reliable starting point for PMTD with only ~1 minute of initialization time, significantly improving training efficiency and final model quality.
>
>
>
> > **Q4:** Computational Analysis of the BTBI Module and other baselines
>
> **A4:** We evaluate the cost time, number of search iterations, and final performance (PSNR) on multiple STVSR benchmarks under both 2-bit and 4-bit settings. As shown in the Table 1, our BTBI method achieves significantly lower runtime compared to other methods such as DOBI(2dquant) and DBDC, while delivering the best overall performance. This efficiency stems from two key design choices:A tightly bounded search space based on percentiles and An early-pruning strategy that avoids unnecessary evaluations.
> Although BTBI involves slightly more iterations (due to per-frame bound search), this is a one-time offline process. During inference, the model uses the precomputed quantization bounds just like other PTQ methods, introducing no additional runtime or memory overhead.
> These results validate the Superiority of our approach: BTBI enables finer-grained per-frame quantization with minimal cost.
>
> #### Table 1: Performance and Computational Cost Comparison of BTBI on STVSR
> | Method      | Bit   | Cost Time  | Iterations | **Vid4**  | **Vimeo-Fast** | **Vimeo-Medium** | **Vimeo-Slow** |
> | ----------- | ----- | ---------- | ---------- | --------- | -------------- | ---------------- | -------------- |
> | RSTT-S (FP) | 32/32 | N/A        | N/A        | 26.29     | 36.58          | 35.43            | 33.30          |
> | Percentile  | 2/2   | N/A        | N/A        | 12.64     | 15.23          | 14.82            | 14.67          |
> | MinMax      | 2/2   | N/A        | N/A        | 10.44     | 10.54          | 10.36            | 10.21          |
> | DOBI        | 2/2   | 118.98s    | 400.00     | 20.15     | 23.19          | 23.33            | 23.59          |
> | DBDC        | 2/2   | 173.37s    | 153.73     | 18.95     | 23.49          | 23.33            | 23.49          |
> | **BTBI**    | 2/2   | **70.89s** | **410.52** | **21.45** | **24.63**      | **25.28**        | **25.96**      |
> | Percentile  | 4/4   | N/A        | N/A        | 23.14     | 27.09          | 26.96            | 26.59          |
> | MinMax      | 4/4   | N/A        | N/A        | 22.67     | 26.45          | 25.87            | 25.47          |
> | DOBI        | 4/4   | 130.17s    | 400.00     | 23.95     | 31.59          | 29.93            | 30.59          |
> | DBDC        | 4/4   | 200.47s    | 167.73     | 23.72     | 30.41          | 29.98            | 30.54          |
> | **BTBI**    | 4/4   | **80.89s** | **421.45** | **24.15** | **32.24**      | **30.77**        | **30.66**      |
>
> > **Q5:** Please clarify the workflow.
>
> **A5:** The workflow of PMQ-VE follows a simple and modular two-stage pipeline:
> 1. Input: A pretrained full-precision (FP32) model.
> 2. Stage 1 — Coarse (BMFQ):
> Perform per-frame clipping bound search using our Backtracking-based Multi-Frame Quantization (BMFQ), which computes robust activation ranges for each frame.
> 3. Stage 2 — Fine (PMTD):
> Apply Progressive Multi-Teacher Distillation (PMTD) to transfer knowledge from full-precision and intermediate-bit (e.g., 8/4-bit) teacher models to the low-bit student model (e.g., 4/2-bit).
> 4. Output: A quantized model with integer weights and fixed activation bounds.
> 5. Deployment and Inference: The model performs efficient integer matrix operations using precomputed quantization parameters, without any additional search or adaptation
>
>
>
> > **Q6:** How are the intermediate-bit teachers trained?
>
> **A6:** Teachers are progressively trained:
> - 8-bit model: Trained using BMFQ + distillation from FP32 supervision.
> - 4-bit model: Trained using BMFQ + distillation from both FP32 and 8-bit supervision.
> - 2-bit model: Trained using BMFQ + distillation from FP32, 8-bit, and 4-bit supervision.
> This hierarchical supervision bridges the capacity gap between FP32 and low-bit models, and is validated through ablation studies (Table 4, Table 6) in our submission.
>
>
> We noticed that several of your comments reflect a misunderstanding of some fundamental concepts underlying our work. Please feel free to reach out if anything remains unclear.
>
> [1] Chen Z, Qin H, Guo Y, et al. Binarized diffusion model for image super-resolution[J]. Advances in Neural Information Processing Systems, 2024, 37: 30651-30669.
>
> [2] Yuan Z, Xue C, Chen Y, et al. Ptq4vit: Post-training quantization for vision transformers with twin uniform quantization[C]//European conference on computer vision. Cham: Springer Nature Switzerland, 2022: 191-207.
>
> [3] Tu Z, Hu J, Chen H, et al. Toward accurate post-training quantization for image super resolution[C]//Proceedings of the IEEE/CVF Conference on Computer Vision and Pattern Recognition. 2023: 5856-5865.

---

> ### Comment · Reviewer_7rvP · 2025-08-02
>
> Thank you for your explanations and experiments. My questions about the computational complexity and intermediate-bit teachers have been addressed. However, I still have many critical questions about BMFQ.
>
> 1. There are two contradictory statements in the rebuttal: “Our approach is Post-Training Quantization (PTQ). … The purpose is to learn quantization parameters for both weights and activations.” and “PTQ keeps weights fixed and only learns clipping bounds.” These two contradictory statements make me confused. Does your approach actually learn the quantized weights?
> 2. Since this is a PTQ method according to your rebuttal, why does the Problem Formulation sub-section in your main text introduce fake quantization and say “adopt STE during training”? (As far as I know, fake quantization is a QAT method.) This may mislead readers to regard your work as a QAT approach.
> 3. Based on my understanding of your manuscript, the term “multi-frame quantization” implies that you are searching for the optimal lb and ub for each frame in the calibration video and saving the parameters. If that is the case, there are three key issues that need to be clarified:
>
> 3.1   What if the number of frames in the calibration video differs from the number of frames in the test video? For example, if the calibration video has only 1000 frames, you can only obtain 1000 groups of lb and ub parameters. However, if the test video has 2000 frames, how would you apply 1000 groups of parameters to 2000 frames?
> 3.2   Let’s still take the 1000-frame calibration video as an example. After quantization, you need to store 1000 groups of parameters. This storage overhead may already be far greater than storing a full-precision (FP32) model. The larger the calibration set, the more parameters you need to store. Have you conducted a quantitative analysis of this issue?
> 3.3  The temporal characteristics of the calibration video and the test video may be completely different. For example, the video used for calibration gradually brightens from extremely dark, while the video used for testing gradually darkens from extremely bright. However, the parameters obtained by overfitting on the calibration video are only applicable to videos with the “gradually brightening from dark” temporal characteristic, and may be completely unsuitable for new test videos.
>
> Please provide explanations for these key issues.

---

> > ### Author Response · Authors · 2025-08-04
> >
> > > **Q1:** Clarification on PTQ Definition
> >
> > **A1:** Our method follows the PTQ paradigm, where model weights remain frozen and only quantization parameters—clipping bounds for weights and activations—are optimized. While quantized weights are used during the forward pass, the underlying full-precision weights are never updated or fine-tuned. This aligns with common PTQ practices in prior work. For example：Section 2.1 of [1] states that "PTQ methods should determine the scaling factors ∆ of activations and weights for each layer."
> > Section 3.3 of [2] states that "the original weight parameters remain unchanged".
> >
> > > **Q2:** Clarification on the Use of STE and Fake Quantization within PTQ
> >
> > **A2:** In PMTD, we refine the clipping bounds via knowledge distillation, using STE and fake quantization to optimize the quantization parameters. The core distinction between PTQ and QAT lies in whether the weights are updated via backpropagation—not in the use of fake quantization or STE (which are also used in PTQ methods [3], [4]).
> >
> > > **Q3.1:** Applicability of Frame-Wise Quantization to Arbitrary-Length Videos
> >
> > **A3.1:** This issue does not occur even with an arbitrary number of input frames, because VSR/STVSR/VFI models process the video in a sliding or recurrent manner, where each inference step involves a fixed number of frames or hidden states. As explained in Section 2.1 of [5], these models fall into two categories:
> > - Sliding-window based methods process a fixed number of frames.
> > - Recurrent-based methods process the current frame along with one-/two-order propagated hidden states from previous steps — also a fixed input size.
> >
> >
> >
> > > **Q3.2:** Efficiency of Clipping Parameter Storage in Practical Scenarios
> >
> > **A3.2:** As clarified in A3.1, our method only requires storing a fixed number of frame-wise bounds. Put the overhead into perspective, consider a single linear layer with 512 input and 512 output dimensions contains 262,144 weights, which takes about 1MB in FP32. In contrast, we only store 2 float32 values (lb and ub) for this weight.
> > For activations, even in a multi-frame setting with N = 7, we only store 2 × N = 14 float32 values (56 bytes). Overall, the storage overhead introduced by our method is negligible compared to the model weights.
> >
> > > **Q3.3:** Generalization Capability of Clipping Bounds Across Temporal Variations
> >
> > **A3.3:** This corner case is inherent to all PTQ methods, not specific to our approach. In practice, our method mitigates this issue by calibrating on a diverse set of videos, which includes a wide range of temporal patterns — brightening, darkening, fast motion, slow motion — to ensure that the learned bounds are globally robust rather than tailored to a narrow temporal trajectory.
> >
> > Our strong performance across various benchmarks (e.g., Vimeo-Fast, Vimeo-Slow) — each with different motion and illumination characteristics — offers empirical evidence that the chosen bounds generalize well under typical conditions. However, in extreme edge cases where calibration and test videos have completely disjoint temporal behaviors, some performance drop is expected.
> >
> > We believe such cases reflect a data representativeness issue during calibration, rather than a flaw in the quantization framework itself. As a practical remedy, our method is lightweight and efficient: the bound search takes ~1 minute, and refinement only ~6 hours, making it practical to re-run the process on new video distributions when necessary.
> >
> > Lastly, we note that similar generalization issues exist in image-level quantization as well — for instance, when calibrating on dark images and testing on bright ones. Yet, with representative data, these methods still perform well in practice. We view this as a common challenge in quantization, not a fundamental weakness of our design.
> >
> > [1] Yuan Z, Xue C, Chen Y, et al. Ptq4vit: Post-training quantization for vision transformers with twin uniform quantization[C]//European conference on computer vision. Cham: Springer Nature Switzerland, 2022: 191-207.
> >
> > [2] Zhu L, Li J, Qin H, et al. Passionsr: Post-training quantization with adaptive scale in one-step diffusion based image super-resolution[C]//Proceedings of the Computer Vision and Pattern Recognition Conference. 2025: 12778-12788.
> >
> > [3] Tu Z, Hu J, Chen H, et al. Toward accurate post-training quantization for image super resolution[C]//Proceedings of the IEEE/CVF Conference on Computer Vision and Pattern Recognition. 2023: 5856-5865.
> >
> > [4] Liu K, Qin H, Guo Y, et al. 2DQuant: Low-bit post-training quantization for image super-resolution[J]. Advances in Neural Information Processing Systems, 2024, 37: 71068-71084.
> >
> > [5] Zhou X, Zhang L, Zhao X, et al. Video super-resolution transformer with masked inter&intra-frame attention[C]//Proceedings of the IEEE/CVF Conference on Computer Vision and Pattern Recognition. 2024: 25399-25408.

---

> > > ### Comment · Reviewer_7rvP · 2025-08-04
> > >
> > > Thank you for your reply. I now have a clearer understanding of the details of your method, and will raise my evaluation score. However, I suggest that the authors significantly improve the clarity of the writing and incorporate some key issues we discussed into the final version. Particularly, the authors should clarify that you are searching for the optimal lb and ub parameters for EACH FRAME IN THE SLIDING WINDOW, rather than EACH FRAME IN THE CALIBRATION VIDEO, so that you can apply your approach to videos of arbitrary temporal length. This crucial point is completely missing in your current manuscript, which is likely to result in misunderstanding.

---

> > > > ### Author Response · Authors · 2025-08-04
> > > >
> > > > Thank you very much for your thoughtful follow-up and for raising your evaluation score. We sincerely appreciate the time and effort you have taken to understand our method in depth.
> > > >
> > > > We will revise the final version to clearly clarify that our method searches for lb and ub for each frame in the sliding window, rather than each frame in the calibration video, which enables generalization to videos of arbitrary length. We will also add a quantitative analysis of storage and computational overhead, and clarify the PTQ setting and the use of STE .
> > > >
> > > > Thank you again for your constructive suggestions — they are instrumental in improving the clarity and quality of our work.

---

### Official Review · Reviewer_LizY · 2025-07-03

**Clarity:** 3
**Significance:** 3
**Originality:** 3
**Rating:** 5
**Confidence:** 4

**Summary:**

The authors propose a multi-frame video enhancement method relying on two underlying concepts - a)Backtracking based quantization - involves frame based adaptive quantization where the quantization bounds are obtained adaptively for every frame using backtracking, b) Progressive multi-teacher distillation - this framework aims to reproduce representational capacity of full-precision teacher by employing a progressive training scheme with intermediate 8/4-bit teacher. Extensive experiments across 3 tasks : Video Frame Interpolation (VFI), Video Super-Resolution (VSR), and Spatio-Temporal Video Super-Resolution (STVSR) are performed. The results show improvement over existing state-of-the-art low-bit models.

**Questions:**

1. Although the backtracking based quantization is proposed for frame based manner from the implementation details it appears to be done at a crop/patch level. Just wondering if spatial dimensions have any impact? Additionally if instead of doing backtracking in a patch-based manner instead of frame level are there any pros/cons associated with it?

2. What is the average recursion depth observed in the training? Can that be reduced by leveraging certain assumptions such as the motion does not change significantly between consecutive frames?

**Ethical Concerns:**

["NO or VERY MINOR ethics concerns only"]

**Final Justification:**

The authors have reasonably addressed the concerns. Since I had a high rating before I am keeping the same decision.

**Limitations:**

As the authors have pointed out, diffusion models aren't included in this work. It will interesting to know how they perform with this technique.

**Quality:**

3

**Strengths And Weaknesses:**

Strengths
1. The idea of adaptively modifying quantization for each frame using backtracking is a novel idea and hasn't been explored earlier. Also from the ablation experiments it is evident that this provides the highest performance boost.

2. Progressive quantization is a simple but powerful idea where distillation is done to very low-bits in a progressive manner. From the results this also leads to better performance.

3. The paper is presented in an elegant manner and the algorithm is easy to comprehend. Overall the proposed technique improves upon the existing state-of-the-art and is generalizable across different video enhancement tasks as well as can be applied across different deep learning models.


Weaknesses
1. The quantization scheme proposed involves a recursive backtracking strategy performed on a frame level manner. This could be compute heavy for videos with large number of frames and/or with high resolution. Also authors haven't reported the computational complexity of their method, so hard to understand how different video characteristics impact on the compute.

2. Although the authors propose the above method for video enhancement there isn't any specific information associated with enhancement (except it is applied for each frame) that is employed in the design. Perhaps the same design can be applicable to other tasks as well? For example in videos there is significant overlap between temporal information between successive frames. This could be employed to reduce the recursive search as doing backtracking based search for every video frame could be wasteful as well as expensive.

---

> ### Author Rebuttal · Authors · 2025-07-31
>
> We appreciate the reviewer’s recognition of our work, particularly the novelty of our frame-wise backtracking quantization strategy, the effectiveness and simplicity of the progressive multi-teacher distillation framework, and the strong empirical results across VFI, VSR, and STVSR tasks. We are also grateful for the positive feedback regarding the clarity and generalizability of our approach.
> > **Q1:** Computational Complexity and Average recursion Iterations of the BTBI Module and other baseline methods
>
> **A1:** Following your advice, we compare the average number of iterations and the running time of our BTBI method with other baseline methods. As shown in Table 1, while our per-frame backtracking strategy introduces more iterations due to per-frame bound search, the actual runtime remains significantly lower than existing search-based approaches(e.g., DOBI from 2DQuant, DBDC). This efficiency is mainly attributed to our percentile-constrained search space and early-pruning design, which effectively narrows down the candidate bounds and avoids unnecessary evaluations.
> Notably, this overhead is only introduced once during the offline quantization phase. During inference, BTBI shares the same runtime cost as other post-training quantization methods, as it simply uses the precomputed quantization parameters without any additional computation, regardless of video length or resolution.
>
> #### Table 1: Performance and Computational Cost Comparison of BTBI on STVSR
> | Method          | Bit   | Cost Time  | Iterations | **Vid4**  | **Vimeo-Fast** | **Vimeo-Medium** | **Vimeo-Slow** |
> | --------------- | ----- | ---------- | ---------- | --------- | -------------- | ---------------- | -------------- |
> | RSTT-S (FP)     | 32/32 | N/A        | N/A        | 26.29     | 36.58          | 35.43            | 33.30          |
> | Percentile      | 2/2   | N/A        | N/A        | 12.64     | 15.23          | 14.82            | 14.67          |
> | MinMax          | 2/2   | N/A        | N/A        | 10.44     | 10.54          | 10.36            | 10.21          |
> | DOBI            | 2/2   | 118.98s    | 400.00     | 20.15     | 23.19          | 23.33            | 23.59          |
> | DBDC            | 2/2   | 173.37s    | 153.73     | 18.95     | 23.49          | 23.33            | 23.49          |
> | **BTBI (Ours)** | 2/2   | **70.89s** | **410.52** | **21.45** | **24.63**      | **25.28**        | **25.96**      |
> | Percentile      | 4/4   | N/A        | N/A        | 23.14     | 27.09          | 26.96            | 26.59          |
> | MinMax          | 4/4   | N/A        | N/A        | 22.67     | 26.45          | 25.87            | 25.47          |
> | DOBI            | 4/4   | 130.17s    | 400.00     | 23.95     | 31.59          | 29.93            | 30.59          |
> | DBDC            | 4/4   | 200.47s    | 167.73     | 23.72     | 30.41          | 29.98            | 30.54          |
> | **BTBI (Ours)** | 4/4   | **80.89s** | **421.45** | **24.15** | **32.24**      | **30.77**        | **30.66**      |
>
>
> > **Q2:** Task-Specificity of PMQ-VE
>
> **A2:** We agree with the reviewer that our method is general and can be applied to other tasks beyond video enhancement.
> In this paper, we focus on three representative video enhancement tasks—VFI, VSR, and STVSR—because they cover the core challenges in video enhancement:
> - VFI captures temporal dependencies,
> - VSR focuses on spatial reconstruction,
> - STVSR requires joint spatio-temporal modeling.
>
> These tasks are not only fundamental but also diverse in their modeling requirements, making them ideal benchmarks to validate the generality and robustness of our quantization framework. Since most other video tasks also rely on spatial and/or temporal cues, our method can be naturally extended to them.
>
> > **Q3:** Whether Spatial Resolution (Patch vs. Frame level) Affects the Quantization Process
>
> **A3:**  Spatial dimensions do not significantly impact the quantization process in our framework. Although we use patch-based augmentation(e.g., cropping) during training, quantization is consistently performed at the frame level. At inference, the learned per-frame clipping bounds are applied to full-resolution frames, and this works well because normalization layers (e.g., LayerNorm) stabilize activation distributions across spatial scales. As a result, whether the input is a full frame or a patch, activations are mapped to a consistent range, ensuring quantization stability.
>
> We do not adopt patch-level quantization because it would require modifying the inference pipeline, adding unnecessary complexity and inconsistency with training.
>
>
> > **Q4:** Leveraging Temporal Redundancy to Reduce Recursive Search Overhead in BTBI
>
> **A4:** Thank you for the insightful suggestion. We agree that consecutive video frames share considerable temporal redundancy, and this can be leveraged to reduce the computational cost of the per-frame recursive bound search in BTBI.
> To address this, we implemented a **temporal warm-start strategy**, where the BTBI search for frame *i* is initialized using the optimal bounds obtained from the previous frame *i−1*.
> Below is a simplified illustration of the idea:
> ```python
> # Temporal warm-start BTBI
> for i in range(T):
>     if i == 0:
>         lb_i, ub_i = BTBI_search(X[0])  # standard search
>     else:
>         lb_i, ub_i = BTBI_search(X[i], init=(lb_{i−1}, ub_{i−1}))  # warm start
> ```
> We compare the original BTBI and the warm-start variant in the table2. As shown, the warm-start version substantially reduces both the average iteration count and runtime cost, while maintaining nearly identical performance in terms of PSNR.
> This warm-start strategy brings consistent improvements in both runtime and iteration number, with negligible impact on final restoration quality. It demonstrates that leveraging temporal continuity is an effective way to reduce BTBI overhead in real video applications.
>
> We sincerely thank the reviewer for this excellent suggestion, which directly helped us improve the efficiency of the BTBI module. We will include this strategy, its implementation, and the empirical results in the final version of the paper.
>
> #### Table 2: Effect of Warm-Start Strategy on BTBI (2-bit and 4-bit Quantization)
> | Method          | Bit | BTBI Cost Time | iterations | **Vid4** | **Fast** | **Medium** | **Slow** |
> | --------------- | --- | -------------- | ---------- | -------- | -------- | ---------- | -------- |
> | BTBI            | 2/2 | 70.89          | 410.52     | 21.45    | 24.63    | 25.28      | 25.96    |
> | BTBI+warm-start | 2/2 | 63.22          | 298.91     | 21.43    | 24.62    | 25.26      | 25.94    |
> | BTBI            | 4/4 | 80.89          | 421.45     | 24.15    | 32.24    | 30.77      | 30.66    |
> | BTBI+warm-start | 4/4 | 72.21          | 312.67     | 24.13    | 32.23    | 30.75      | 30.65    |
>
> We greatly appreciate the reviewer’s thoughtful and constructive feedback. All raised concerns—including BTBI’s computational overhead, task generality, spatial resolution effects, and the suggestion to leverage temporal redundancy—will be further clarified in the final version to strengthen the technical clarity and  quanlity of our work.

---

> > ### Comment · Reviewer_LizY · 2025-08-07
> > **Addressing Concerns**
> >
> > The authors have addressed the concerns raised during the review. Since my original decision had a higher evaluation score, I am keeping the same score.

---

### Official Review · Reviewer_wyAz · 2025-07-05

**Clarity:** 3
**Significance:** 3
**Originality:** 3
**Rating:** 4
**Confidence:** 3

**Summary:**

Transformer-based video enhancement models are too resource-intensive for edge deployment, while traditional quantization methods fail because they cannot handle varying data distributions across frames or the large representation gap between teacher and student models. To solve this, this paper proposes a novel two-stage method PMQ-VE, containing the Backtracking-based Multi-Frame Quantization (BMFQ) and the Progressive Multi-Teacher Distillation (PMTD). Experiments show that PMQ-VE achieves state-of-the-art performance on multiple benchmarks.

**Questions:**

1. Could you further analyze and discuss the computational overhead of the BTBI module, particularly in comparison to baseline methods?

2. Could you comment on the stability and reliability of the results, and elaborate on why the substantial performance gains are reported from a single run?

3. Could you elaborate on the selection of the new hyperparameters and discuss the expected sensitivity of the framework's performance to their variations?

**Ethical Concerns:**

["NO or VERY MINOR ethics concerns only"]

**Limitations:**

yes

**Quality:**

3

**Strengths And Weaknesses:**

Strengths:

1. This paper investigates and solves the problem of quantizing multi-frame video enhancement models. This is a practical and important problem, as the efficient deployment of video models is a key bottleneck for many real-world applications.

2. The proposed PMQ-VE framework has two core components, BMFQ and PMTD, effectively addressing the corresponding challenges.

3. The results consistently demonstrate the superiority of the proposed method, with significant gains in PSNR/SSIM on multiple datasets.

4. The paper is well-written and easy to understand.

Weaknesses:

1. The paper lacks detailed quantitative analysis and proof of the computational cost of proposed modules such as BTBI, and lacks comparison with other baseline methods.

2. All the experimental results reported in this paper are the results of a single experiment, rather than the standard deviations and means of multiple experiments. While the performance gains are substantial, reporting statistics over multiple independent experiments can demonstrate the stability and reliability of the results.

3. The proposed framework introduces new hyperparameters. However, the paper does not provide a sensitivity analysis for these parameters, making it difficult to assess the method's robustness or how to tune them for new scenarios.

---

> ### Author Rebuttal · Authors · 2025-07-31
>
> We sincerely appreciate the reviewers' recognition of the novelty, practicality, and effectiveness of our proposed PMQ-VE framework for quantization in multi-frame video enhancement, as well as its strong performance across various benchmarks and the clarity of our writing.
>
> > **Q1:** Lack of quantitative analysis and comparison of computational cost for BTBI and baseline methods
>
> **A1:** Following your advice, we evaluate the cost time, number of search iterations, and final performance (PSNR) on multiple STVSR benchmarks under both 2-bit and 4-bit settings. As shown in Table 1, our BTBI method achieves significantly lower running time compared to other search-based methods (e.g., DOBI from 2DQuant, DBDC), while achieveing the best overall performance.
> This efficiency stems from two key design: (1) a tightly bounded search space based on percentiles, and (2) an early-pruning mechanism that eliminates redundant evaluations. Although BTBI performs more iterations due to per-frame bound search, this is a one-time offline process. During inference, the model uses precomputed bounds like all other PTQ methods, incurring no additional running time or memory overhead.
>
>
> #### Table 1: Performance and Computational Cost Comparison of BTBI on STVSR (5-run average ± std)
> | Method          | Bit   | Cost Time       | iterations      | **Vid4**       | **Vimeo-Fast** | **Vimeo-Medium** | **Vimeo-Slow** |
> | --------------- | ----- | --------------- | --------------- | -------------- | -------------- | ---------------- | -------------- |
> | RSTT-S(FP)      | 32/32 | N/A             | N/A             | 26.29          | 36.58          | 35.43            | 33.30          |
> | Percentile      | 2/2   | N/A             | N/A             | 12.64±0.29     | 15.23±0.24     | 14.82±0.18       | 14.67±0.21     |
> | MinMax          | 2/2   | N/A             | N/A             | 10.44±1.19     | 10.54±0.25     | 10.36±0.16       | 10.21±0.41     |
> | DOBI            | 2/2   | 118.98s±1.23s   | 400.00±0.00     | 20.15±0.14     | 23.19±0.13     | 23.33±0.09       | 23.59±0.18     |
> | DBDC            | 2/2   | 173.37±1.72s    | 153.73±20.47    | 18.95±0.35     | 23.49±0.14     | 23.33±0.11       | 23.49±0.19     |
> | **BTBI (Ours)** | 2/2   | **70.89±0.57s** | **410.52±8.42** | **21.45±0.12** | **24.63±0.12** | **25.28±0.12**   | **25.96±0.20** |
> | Percentile      | 4/4   | N/A             | N/A             | 23.14±0.21     | 27.09±0.11     | 26.96±0.34       | 26.59±0.51     |
> | MinMax          | 4/4   | N/A             | N/A             | 22.67±0.68     | 26.45±0.13     | 25.87±0.46       | 25.47±0.51     |
> | DOBI            | 4/4   | 130.17s±7.23s   | 400.00±0.00     | 23.95±0.12     | 31.59±0.05     | 29.93±0.31       | 30.59±0.31     |
> | DBDC            | 4/4   | 200.47±30.72s   | 167.73±18.96    | 23.72±0.19     | 30.41±0.15     | 29.98±0.24       | 30.54±0.51     |
> | **BTBI (Ours)** | 4/4   | **80.89±1.98s** | **421.45±7.31** | **24.15±0.11** | **32.24±0.12** | **30.77±0.23**   | **30.66±0.21** |
>
>
> > **Q2:** Lack of statistical reporting across multiple runs; concerns about stability and reproducibility
>
> **A2:** As shown in Tables 2, 3, and 4, we conduct five independent runs of our method on all three tasks (STVSR, VFI, and VSR), each with a different random seed and calibration set. The resulting PSNR values and their standard deviations (typically < ±0.1 dB) demonstrate the stability, robustness, and reproducibility of our method under low-bit quantization.
>
> #### Table 2: STVSR (5-run average ± std)
> | Bit | Vid4         |                 | Vimeo-Fast   |                 | Vimeo-Medium |                 | Vimeo-Slow   |                 |
> | --- | ------------ | --------------- | ------------ | --------------- | ------------ | --------------- | ------------ | --------------- |
> |     | PSNR         | SSIM            | PSNR         | SSIM            | PSNR         | SSIM            | PSNR         | SSIM            |
> | 2/2 | 23.45 ± 0.06 | 0.6214 ± 0.0014 | 30.34 ± 0.05 | 0.8442 ± 0.0013 | 30.17 ± 0.06 | 0.8543 ± 0.0013 | 29.11 ± 0.06 | 0.8315 ± 0.0014 |
> | 4/4 | 25.42 ± 0.05 | 0.7542 ± 0.0023 | 34.66 ± 0.08 | 0.9165 ± 0.0012 | 33.76 ± 0.07 | 0.9117 ± 0.0023 | 31.93 ± 0.06 | 0.8938 ± 0.0013 |
>
> #### Table 3: VFI (5-run average ± std)
> | Bit | EMA-VFI [T]  |                 |               |             | EMA-VFI [D]  |                 |               |             |
> | --- | ------------ | --------------- | ------------- | ----------- | ------------ | --------------- | ------------- | ----------- |
> |     | PSNR         | SSIM            | LPIPS         | NIQE        | PSNR         | SSIM            | LPIPS         | NIQE        |
> | 4/4 | 28.41 ± 0.04 | 0.9152 ± 0.0022 | 0.131 ± 0.003 | 7.34 ± 0.05 | 29.57 ± 0.03 | 0.9362 ± 0.0012 | 0.101 ± 0.002 | 6.74 ± 0.03 |
>
> #### Table 4: VSR (5-run average ± std)
> | Bit | Vimeo90K     |                 | Vid4         |                 |
> | --- | ------------ | --------------- | ------------ | --------------- |
> |     | PSNR         | SSIM            | PSNR         | SSIM            |
> | 4/4 | 37.32 ± 0.05 | 0.9442 ± 0.0012 | 27.65 ± 0.04 | 0.8350 ± 0.0020 |
>
> > **Q3:** Sensitivity of newly introduced hyperparameters and their impact on robustness
>
> **A3:** We conducted a sensitivity analysis on the key hyperparameters of PMQ-VE, including:
> - The distillation loss weight λ
> - The number of BTBI search steps N (which controls ΔL/ΔU)
> - The early-stopping threshold ε
> - The percentile search ranges for `lb` and `ub`
> We varied each parameter individually while keeping the others fixed, and evaluated the resulting performance under 4-bit quantization. As shown in Table 5, across all variations, the PSNR remains within ±0.1 dB of the default setting, demonstrating that our method is robust to reasonable changes in hyperparameter configurations. These minimal fluctuations confirm that our coarse-to-fine framework is robust to reasonable changes in hyperparameter configurations.
>
> #### Table 5: Hyperparameter Sensitivity Analysis Under 4-bit Quantization (5-run average ± std)
> | Variant          | N   | ε    | Range                    | λ   | BTBI Time   | BTBI Iters   | **Vid4**   | **Vimeo-Fast** | **Vimeo-Medium** | **Vimeo-Slow** |
> | ---------------- | --- | ---- | ------------------------ | --- | ----------- | ------------ | ---------- | -------------- | ---------------- | -------------- |
> | **Default**      | 10  | 5e-4 | [0.1, 10] / [90, 99.9]   | 5   | 80.89±1.98s | 421.45±7.31  | 25.42±0.02 | 34.66±0.03     | 33.76±0.05       | 31.93±0.04     |
> | N ↑              | 11  | 5e-4 | [0.1, 10] / [90, 99.9]   | 5   | 81.45±2.45s | 432.46±8.54  | 25.42±0.08 | 34.68±0.05     | 33.79±0.04       | 31.92±0.06     |
> | N ↓              | 9   | 5e-4 | [0.1, 10] / [90, 99.9]   | 5   | 80.23±3.44  | 420.31±8.91  | 25.41±0.05 | 34.66±0.06     | 33.75±0.07       | 31.90±0.01     |
> | ε ↑              | 10  | 6e-4 | [0.1, 10] / [90, 99.9]   | 5   | 81.35±2.64  | 432.55±6.88  | 25.40±0.04 | 34.65±0.06     | 33.74±0.06       | 31.92±0.04     |
> | ε ↓              | 10  | 4e-4 | [0.1, 10] / [90, 99.9]   | 5   | 79.91±3.41  | 401.75±8.79  | 25.41±0.04 | 34.67±0.04     | 33.75±0.04       | 31.92±0.03     |
> | Wider Range ↑    | 10  | 5e-4 | [0.01, 15] / [85, 99.99] | 5   | 83.89±2.54  | 434.65±6.45  | 25.38±0.03 | 34.61±0.06     | 33.71±0.04       | 31.89±0.02     |
> | Narrower Range ↓ | 10  | 5e-4 | [0.1, 1] / [99, 99.9]    | 5   | 82.84±12.54 | 430.22±30.45 | 25.41±0.06 | 34.59±0.03     | 33.69±0.04       | 31.90±0.02     |
> | λ ↑              | 10  | 5e-4 | [0.1, 10] / [90, 99.9]   | 5.5 | 80.89±5.54  | 421.45±10.41 | 25.42±0.01 | 34.67±0.06     | 33.73±0.03       | 31.93±0.05     |
> | λ ↓              | 10  | 5e-4 | [0.1, 10] / [90, 99.9]   | 4.5 | 80.89±3.48  | 421.45±9.10  | 25.41±0.06 | 34.66±0.05     | 33.75±0.04       | 31.93±0.05     |
>
>
> We greatly appreciate the reviewer’s valuable suggestions. All raised concerns—including computational analysis, statistical reporting, and hyperparameter sensitivity—will be addressed in the final version to enhance the completeness and clarity of our submission.

---

### Comment · Area_Chair_SwEX · 2025-08-04
**rebuttal discussion**

Just a quick reminder to please read the author rebuttals for your assigned papers. If you have any remaining questions or concerns, feel free to initiate the discussion to help guide a constructive decision-making process. Your input is important and much appreciated!

---

### Note · Authors · 2025-08-12

We sincerely thank all reviewers and the AC for their time and constructive feedback. We are encouraged that all reviewers recognized the practical value, novelty, and empirical strength of our proposed PMQ-VE framework. In particular:

Reviewer wyAz appreciated the clarity and consistent performance of our method across tasks.

Reviewer LizY highlighted the novelty of our frame-wise backtracking quantization and progressive distillation.

Reviewer B6R2 acknowledged the comprehensiveness of our experiments and the efficiency of our quantization design.

Notably, Reviewer 7rvP, who initially gave a lower score (3), raised several concerns regarding the PTQ/QAT distinction, per-frame bound generalization, and the computational overhead of our BTBI module. We carefully clarified these issues, including:

- Our method is strictly PTQ, with no weight updates.
- Per-frame bounds are applied over sliding windows, not fixed calibration sequences, enabling generalization to arbitrary-length videos.
- Storage and runtime overheads are negligible and we provided quantitative analysis to support this.

Following our response, Reviewer 7rvP explicitly acknowledged the clarifications and increased score, and Reviewer B6R2 did the same after our additional experiments and analysis.

We will incorporate these clarifications into the final version to enhance the clarity and quality of our work. Once again, we thank all reviewers and the AC for their thoughtful reviews and helpful discussions throughout this process.

---

### Decision · Program_Chairs · 2025-09-17

**Decision:**

Accept (poster)

**Comment:**

The paper received BA, A, BA, and A ratings. Overall, the reviewers have found some merit in this work.